# Adaptive Batch-Wise Sample Scheduling for Direct Preference Optimization

**Zixuan Huang**[1]    **Yikun Ban**[1*]  **Lean Fu**[2]   **Xiaojie Li**[1]
**Zhongxiang Dai**[3]    **Jianxin Li**[1]    **Deqing Wang**[1*]
[1]Beihang University    [2]Bytedance Inc    [3]The Chinese University of Hong Kong, Shenzhen
{huang_zx, yikunb, li_xiaojie, dqwang}@buaa.edu.cn
lijx@act.buaa.edu.cn  fulean@bytedance.com  daizhongxiang@cuhk.edu.cn

## Abstract

Direct Preference Optimization (DPO) has emerged as an effective approach for aligning large language models (LLMs) with human preferences. However, its performance is highly dependent on the quality of the underlying human preference data. To address this bottleneck, prior work has explored various data selection strategies, but these methods often overlook the impact of the evolving states of the language model during the optimization process. In this paper, we introduce a novel problem: Sample Scheduling for DPO, which aims to dynamically and adaptively schedule training samples based on the model's evolving batch-wise states throughout preference optimization. To solve this problem, we propose SamS, an efficient and effective algorithm that adaptively selects samples in each training batch based on the LLM's learning feedback to maximize the potential generalization performance. Notably, without modifying the core DPO algorithm, simply integrating SamS significantly improves performance across tasks, with minimal additional computational overhead. This work points to a promising new direction for improving LLM alignment through batch-wise sample selection, with potential generalization to RLHF and broader supervised learning paradigms. The code is available at `https://github.com/hzx122/SamS`.

## 1  Introdcution

Direct Preference Optimization (DPO) [69] was proposed as a simpler and more stable alternative to Reinforcement Learning from Human Feedback (RLHF) [20, 102, 64, 38, 17]. As an off-policy preference optimization method, DPO does not require first training an explicit reward model. Instead, given a preference dataset where each sample includes a prompt and a pair of generations with the first one more consistent with human preferences, it directly optimizes a straightforward binary cross-entropy-type objective, which increases the likelihood of chosen response and decreases the likelihood of rejected response. The promise of this approach is that it implicitly optimizes the same objective as RLHF without adding complexity.

Although DPO has demonstrated exceptional performance across a wide range of tasks, its heavy reliance on high-quality human preference data poses a significant bottleneck for practical deployment due to the associated annotation costs. To mitigate this challenge, substantial research efforts have been devoted to enhancing the data quality and utilization in preference optimization. These efforts generally fall into three categories: (1) Active Querying [24, 61, 45]: selecting informative samples for human feedback collection; (2) Response Pair Selection [59, 57]: actively choosing response pairs to annotate conditioned on a given query; (3) Data Pre-selection [73, 25, 34]: identifying and

---

*Deqing Wang and Yikun Ban are corresponding authors.

filtering high-quality samples prior to DPO training. However, approaches in categories (1) and (2) typically only focus on online feedback collection and ignore data quality, while methods in category (3) overlook the evolving internal states of the language model throughout the DPO process.

In contrast to these existing studies, this paper introduces a novel problem: Sample Scheduling for DPO. Specifically, given a fixed preference dataset, the goal is to dynamically and adaptively schedule training samples based on the evolving internal states of the language model during preference optimization. This formulation is motivated by two key challenges: First, as shown in Figure 1a, samples in the training dataset may exhibit varying levels of learning difficulty for different model states. As the models internal state evolves over time, the relative difficulty of each sample may also shift. Without an adaptive scheduling mechanism, the model may overemphasize samples misaligned with its current learning capacity or overfit to some error patterns, thereby impairing its alignment performance [34, 103]. Second, the dataset may contain noisy samples [73]. As shown in Figure 1b, incorrect or inconsistent preference labels can destabilize the DPO training process [36], and low-quality but preferred responses may erode the original conversational ability of the model. We also empirically verify the presence of such noise in Appendix E.

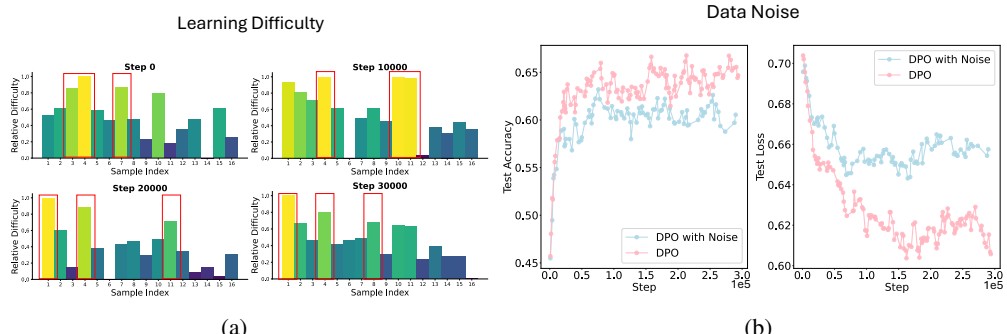

(a)                                                                 (b)

Figure 1: The Study of challenges in SamS. (a) **Varying learning difficulties for different model states.** For the same 16 samples, we track their DPO loss across different states of the language model, from training step 0 to step 30,000. We use the relative DPO loss of each sample as the difficulty measure [34]. (b) **Noisy data degrades DPO performance.** During preference optimization using Pythia-2.8B [14] on the Anthropic-HH dataset [7], we artificially injected 20% noise into the preference labels. As a result, the performance of DPO dropped significantly, highlighting its sensitivity to data quality.

To address this problem, we propose a scheduling algorithm **SamS**, **Sam**ple **S**cheduling for Direct Preference Optimization. In particular, we formulate Sample Scheduling for DPO as a contextual bandit problem, where we define the reward for sample scheduling by leveraging the loss signal during DPO training, and define the arm context based on the internal state representation of LLMs. In this setting, SamS employs a scheduler model to adaptively select samples from each training batch according to the model's evolving states, in order to maximize the potential resulting generalization performance. It incorporates two key innovations. First, it adopts a lagged training strategy, where the scheduler is updated in the subsequent training round, allowing the reward to be collected without incurring additional computational overhead. Second, it introduces an auxiliary exploration network to explicitly address the exploration-exploitation dilemma that is inherent in the iterative sample scheduling problem.

We conduct extensive experiments across diverse benchmarks, including AlpacaEval 2 [29] and MT-Bench [100], to evaluate the effectiveness of SamS. Notably, when integrated with the original DPO loss, SamS consistently outperforms several advanced offline preference optimization methods on mainstream evaluation benchmarks. Particularly, our method improves the AlpacaEval 2 win rate (WR) by 3.0% - 12.4% and the length-controlled win rate (LC) by 5.5% - 8.4% compared to the baselines. Furthermore, we conduct a thorough evaluation of SamS under noisy preference data conditions and show that its integration significantly enhances robustness against label noise. Importantly, thanks to the carefully designed scheduling reward and the lightweight architecture of SamS, the added training overhead is minimal, and GPU memory consumption is even reduced.

In summary, our contributions can be summarized as follows: (1) **Novel Problem**: We introduce a new problem, Sample Scheduling for DPO, which highlights a promising direction for improving LLM alignment performance using fixed preference datasets. (2) **Proposed Algorithm**: We propose SamS, a scheduling algorithm that adaptively selects training samples from each batch according to the model's evolving internal states. (3) **Empirical Effectiveness**: SamS can be seamlessly integrated into existing DPO pipelines without modifications to the core algorithm, yielding substantial performance improvements with only marginal additional computational overhead. Batch-wise sample selection opens a promising path for efficient LLM alignment, and the idea naturally extends to RLHF and other supervised learning paradigms.

## 2 Preliminary

DPO [69] is an offline preference optimization algorithm designed to simplify and stabilize training by reparameterizing the reward function typically used in RLHF. Specifically, DPO reparameterizes the reward model using a closed-form expression:

$$r(x, y) = \beta \log \frac{\pi_\theta(y|x)}{\pi_{\text{ref}}(y|x)} + \beta \log Z(x), \tag{1}$$

where $\pi_\theta$ represents the policy model, $\pi_{\text{ref}}$ is the supervised fine-tuned reference policy, and $Z(x)$ denotes the partition function.

Given a data sample $a = (x, y^w, y^l)$, where $y^w$ and $y^l$ represent the preferred and dispreferred completions respectively for the prompt $x$, the DPO framework incorporates this reward formulation into the Bradley-Terry ranking objective [15]. Specifically, it defines the probability $p(y^w > y^l|x) = \sigma(r(x, y^w) - r(x, y^l))$, where $\sigma$ denotes the logistic function. Consequently, the objective of DPO is formally defined as:

$$\mathcal{L}_{\text{DPO}}(a; \theta) = -\mathbb{E}_{(x, y^w, y^l) \sim \mathcal{D}} \left[ \log \sigma \left( \beta \left( \log \frac{\pi_\theta(y^w|x)}{\pi_{\text{ref}}(y^w|x)} - \log \frac{\pi_\theta(y^l|x)}{\pi_{\text{ref}}(y^l|x)} \right) \right) \right]. \tag{2}$$

In practice, batch-level preference optimization is commonly employed. Given a batch consisting of $n$ samples, denoted as $X_t = \{a_{t,i}\}_{i=1}^n$, where each sample $a_{t,i} = (x_{t,i}, y_{t,i}^w, y_{t,i}^l)$, the average-based DPO loss is formally defined as:

$$\mathcal{L}_{\text{DPO}}(X_t; \theta) = \frac{1}{|X_t|} \sum_{a_{t,i} \in X_t} \mathcal{L}_{\text{DPO}}(a_{t,i}; \theta). \tag{3}$$

During each training round $t \in [T]$, the policy $\pi_\theta$ typically learns from the entire current batch $X_t$, which may contain irrelevant, challenging, or noisy samples. To address this, our objective is to train a scheduler capable of effectively exploring the sample space, thereby identifying and selecting reliable, high-quality samples for the policy's offline preference optimization.

## 3 The Sample Scheduling Problem

We formulate the Sample Scheduling problem for offline preference optimization using the contextual bandit framework proposed in [8, 11, 44]. Let $\pi_\theta$ denote a language model parameterized by $\theta$ that we aim to align with human preferences, and let $f$ denote a scheduler designed to perform interactive sample scheduling during batch-level preference optimization.

**Problem Formulation.** Assume the learning process spans $T$ rounds. At each round $t \in [T]$, we draw a batch containing $n$ samples, denoted by $X_t = \{a_{t,1}, a_{t,2}, \ldots, a_{t,n}\} \sim \mathcal{D}$, where each sample $a_{t,i} = (x_{t,i}, y_{t,i}^w, y_{t,i}^l)$ for $i \in [n]$ is considered an arm, resulting in $n$ total arms. For each arm $a_{t,i}$, we define a contextual representation $\bar{x}_{t,i} = h(x_{t,i}, y_{t,i}^w, y_{t,i}^l)$, where $h(\cdot)$ is an encoding function mapping each sample to a context representation vector.

Given a subset $\widetilde{X}_t \subset X_t$ with size $K$, $|\widetilde{X}_t| = k$, selected by the scheduler $f$, we train the policy $\pi_{\theta_{t-1}}$ on this subset, updating the policy parameters to $\theta_t$ as follows:

$$\theta_t = \theta_{t-1} - \eta \nabla_{\theta_{t-1}} \mathcal{L}_{\text{DPO}}(\widetilde{X}_t; \theta_{t-1}). \tag{4}$$

To measure the improvement from $\theta_{t-1}$ to $\theta_t$ using the selected subset $\widetilde{X}_t$, we introduce a reward function $r(\widetilde{X}_t, \theta_{t-1} \rightarrow \theta_t)$, which is initially unknown. In each round $t \in [T]$, the scheduler $f$ selects a subset $\widetilde{X}_t$ from batch $X_t$ and provides it to policy $\pi_{\theta_{t-1}}$. Subsequently, the scheduler observes the reward $r(\widetilde{X}_t, \theta_{t-1} \rightarrow \theta_t)$, which informs updates to its parameters for future scheduling optimization. The objective for the scheduler $f$ over $T$ rounds is thus to select a sequence of subsets $\{\widetilde{X}_1, \widetilde{X}_2, \ldots, \widetilde{X}_T\}$ that maximizes the cumulative reward:

$$\max \sum_{t=1}^{T} r(\widetilde{X}_t, \theta_{t-1} \rightarrow \theta_t). \tag{5}$$

**Reward Definition.** In supervised preference learning, accurately measuring the performance improvement of policy $\pi_\theta$ from $\theta_{t-1}$ to $\theta_t$ using an oracle is typically impractical. To address this limitation, we propose a reward definition $r$ leveraging insightful information from the learning trajectory of $\theta$. This reward acts as the supervisory signal for training the scheduler $f$ and comprises two distinct components: a batch-level reward and a sample-level reward.

First, we introduce the batch-level reward, which measures the reduction in the average DPO loss before and after training with a selected batch. In practice, we use the batch-average DPO loss to approximate the expected DPO loss across the entire data distribution. Formally, at round $t$, given the policy parameters $\theta_{t-1}$, we train on a subset $\widetilde{X}_t$, resulting in updated parameters $\theta_t$. The batch-level reward for selecting $\widetilde{X}_t$ is defined as:

$$r^B(X_t, \theta_{t-1}, X_{t+1}, \theta_t) = \frac{\overbrace{\sum_{i=1}^{n} e^{\mathcal{L}_{\text{DPO}}(a_{t,i};\theta_{t-1})}}^{A} - \overbrace{\sum_{i=1}^{n} e^{\mathcal{L}_{\text{DPO}}(a_{t+1,i};\theta_t)}}^{B}}{\max\left(\sum_{i=1}^{n} e^{\mathcal{L}_{\text{DPO}}(a_{t,i};\theta_{t-1})}, \sum_{i=1}^{n} e^{\mathcal{L}_{\text{DPO}}(a_{t+1,i};\theta_t)}\right)}. \tag{6}$$

Term $A$ evaluates the performance of $\theta_{t-1}$ on batch $X_t$, noting that $\theta_{t-1}$ has not previously encountered $X_t$. Similarly, term $B$ evaluates the performance of $\theta_t$ on the new batch $X_{t+1}$, after $\theta_t$ has been trained on $\widetilde{X}_t$. To enhance the sensitivity of the reward metric, we exponentiate the DPO loss $e^{\mathcal{L}_{\text{DPO}}(\cdot)}$ and apply normalization through the denominator. Consequently, $r^B$ signifies the approximate performance improvement of the policy $\pi$ after the scheduling decision at round $t$.

Next, we introduce a sample-level reward for fine-grained evaluation, complementing the batch-level reward, which only reflects aggregate improvement over $\widetilde{X}_t$. We assign higher rewards to samples with larger preference margins and greater model uncertainty.

Formally, for a data point $a_{t,i} = (x_{t,i}, y_{t,i}^w, y_{t,i}^l)$, we define the sample-level reward:

$$r^S(a_{t,i}, \theta_{t-1}) = \underbrace{g\left(\beta \log \frac{\pi_{\theta_{t-1}}(y_{t,i}^w \mid x_{t,i})}{\pi_{\text{ref}}(y_{t,i}^w \mid x_{t,i})} - \beta \log \frac{\pi_{\theta_{t-1}}(y_{t,i}^l \mid x_{t,i})}{\pi_{\text{ref}}(y_{t,i}^l \mid x_{t,i})}\right)}_{\text{preference margin}} + \underbrace{\left(1 - g(\log \pi_{\theta_{t-1}}(y_{t,i}^w \mid x_{t,i}))\right)}_{\text{model uncertainty}}. \tag{7}$$

The first term rewards samples with larger preference margins under $\theta_{t-1}$, thereby avoiding convergence on ambiguous or noisy examples [60]. The second term promotes selection of high-uncertainty samples, addressing the tendency of policies to produce out-of-distribution outputs during training [53, 91]. Here, $g(\cdot)$ is a min-max normalization mapping values to $[0, 1]$. We provide a more detailed discussion of the design motivations in the Appendix F.1.

Finally, we integrate both the batch-level and sample-level rewards to compute the final reward for each individual sample $a_{t,i}$ as follows:

$$r(a_{t,i}, \theta_{t-1} \rightarrow \theta_t) = \gamma\sigma\left[r^B(X_t, \theta_{t-1}, X_{t+1}, \theta_t)\right] + (1 - \gamma)\sigma\left[r^S(a_{t,i}, \theta_{t-1})\right], \tag{8}$$

where $\gamma \in [0, 1]$ is a hyperparameter that controls the trade-off between the batch-level and sample-level reward signals, and $\sigma(\cdot)$ denotes the sigmoid function.

To mitigate the combinatorial complexity associated with subset selection, we define the reward of a selected subset $\widetilde{X}_t$ as the sum of the individual sample rewards:

$$r(\widetilde{X}_t, \theta_{t-1} \rightarrow \theta_t) = \sum_{a_{t,i} \in \widetilde{X}_t} r(a_{t,i}, \theta_{t-1} \rightarrow \theta_t). \tag{9}$$

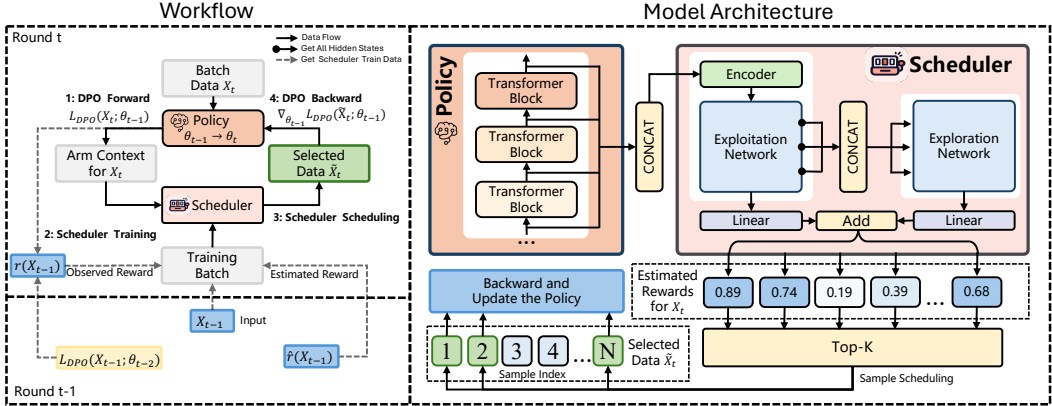

Figure 2: (Left side) Overview of a standard DPO framework integrated with SamS. (Right side) The architecture of the Scheduler. The Scheduler initially treats the policy's hidden state sequence as the arm context for each sample. The Encoder aggregates the state information of each sample to encode the arm context. Subsequently, the Exploitation-Exploration Network utilizes the encoded arm contexts to estimate reward values for each sample, which is used to select a Top-K subset for policy learning.

In addition to providing valuable insights, the reward can be computed straightforwardly during the DPO process.

**Arm Context Design.** The arm context serves as the input to the scheduling module, with the goal of leveraging the representational capacity of the policy model $\pi_\theta$. For each sample, we extract the intermediate hidden representations from all transformer block layers of $\pi_\theta$, and define the arm context as $\bar{x}_{t,i} = h(x_{t,i}, y_{t,i}^w, y_{t,i}^l)$. To obtain a fixed-dimensional vector, we apply a combination of concatenation and pooling operations over the token-level hidden states across layers. This design allows us to take into account the evolution of the state of the LLM into the arm representations.

## 4 Proposed Algorithm: SamS

We present the overall framework of the proposed method, **SamS** (**Sam**ple **S**cheduling), followed by a detailed description of its model structure and workflow. Sample Scheduling can be naturally considered as a sequential decision-making problem under uncertainty, where the internal states of the policy $\theta$ evolve across rounds, resulting in uncertainties for the iterative sample selection. Consequently, the exploration-exploitation dilemma is inherently embedded in this problem.

**Model Structure.** As shown in Figure 2, the scheduler $f$ consists of an encoder layer followed by two specialized networks for exploitation and exploration. The Encoder Layer takes the hidden state representations of each sample as input and produces an encoded representation used by the subsequent neural networks. For notational simplicity, we continue to denote the encoded arm context as $\bar{x}_{t,i}$.

Denote the exploitation network by $f^S(\cdot; \theta^S)$ and the exploration network by $f^{S\prime}(\cdot; \theta^{S\prime})$. The exploitation network $f^S$ learns to predict the reward of each sample arm by mapping the arm context $\bar{x}_{t,i}$ to its observed reward $r(a_{t,i}, \theta_{t-1} \to \theta_t)$. The exploration network $f^{S\prime}$ estimates the uncertainty of the predictions made by $f^S$, and augments the original reward estimate with a potential exploration bonus. This design enables a principled trade-off between exploitation and exploration during iterative sample selection, referring to the design in [8, 12]. This process aligns with the principles of classic Exploration-Exploitation algorithms, such as Upper Confidence Bound (UCB) [9, 10, 67] and Thompson Sampling (TS) [84, 95].

Given the input $\bar{x}_{t,i}$ in round $t$, the exploitation network $f^S$ is implemented as a fully connected feedforward neural network with residual connections, denoted by $f^S(\bar{x}_{t,i}; \theta_t^S)$. After receiving the observed reward $r(a_{t,i}, \theta_{t-1} \to \theta_t)$ in round $t+1$, the parameters $\theta_t^S$ are updated via stochastic

---

**Algorithm 1** Proposed Algorithm: SamS

---

**Require:** $T, n, K, \theta$ (LLM Parameters), $\theta^S, \theta^{S'}$ (Scheduler Parameters)

1: Initialize $\theta_0, \theta_1^S, \theta_1^{S'}$
2: **for** $t = 2, 3, \ldots, T$ **do**
3:     Draw batch data $X_t \sim \mathcal{D}$
    $\nabla$ DPO Forward with $X_t$
4:     Compute DPO Loss $L_{\text{DPO}}(X_t; \theta_{t-1})$  # Standard Forward Pass
    $\nabla$ Scheduler Training
5:     Compute $r^B(X_{t-1}, \theta_{t-2}, X_t, \theta_{t-1})$ based Eq.(6)  # Observe Batch-level Reward
6:     **for** $a_{t-1,i} \in \widetilde{X}_{t-1}$ **do**
7:         Compute $r^S(a_{t-1,i}, \theta_{t-2})$ according to Eq.(7)  # Observe Sample-level Reward
8:         Compute $r(a_{t-1,i}, \theta_{t-2} \to \theta_{t-1})$ according to Eq.(8)  # Observe Final Reward
9:     **end for**
10:    Compute $\mathcal{L}^S(\widetilde{X}_{t-1}, \theta_{t-1}^S)$ According to Eq.(10)
11:    $\theta_t^S = \theta_{t-1}^S - \eta_1 \nabla_{\theta_{t-1}^S} \mathcal{L}^S(\widetilde{X}_{t-1}, \theta_{t-1}^S)$  # Update Exploitation Network of Scheduler
12:    Compute $\mathcal{L}^{S'}(\widetilde{X}_{t-1}, \theta_{t-1}^{S'})$ According to Eq.(11)
13:    $\theta_t^{S'} = \theta_{t-1}^{S'} - \eta_2 \nabla_{\theta_{t-1}^{S'}} \mathcal{L}^{S'}(\widetilde{X}_{t-1}, \theta_{t-1}^{S'})$  # Update Exploration Network of Scheduler
    $\nabla$ Scheduler Scheduling
14:    **for** $i \in [n]$ **do**
15:       $\hat{r}(a_{t,i}, \theta_{t-1} \to \theta_t) = f^S(\bar{x}_{t,i}; \theta_t^S) + \lambda f^{S'}(h_{t,i}^S; \theta_t^{S'})$  # Estimated Reward for Each Sample Based on Exploitation-Exploration Trade-off
16:    **end for**
17:    $\widetilde{X}_t = \text{Top-}K_{i \in [n]} \hat{r}(a_{t,i}, \theta_{t-1} \to \theta_t)$  # Choose $\widetilde{X}_t$
    $\nabla$ DPO Backward with $\widetilde{X}_t$
18:    $\theta_t = \theta_{t-1} - \eta \nabla_{\theta_{t-1}} \mathcal{L}_{\text{DPO}}(\widetilde{X}_t; \theta_{t-1})$  # Udpate LLM with $\widetilde{X}_t$
19: **end for**
20: **Return:** $\theta_T$

---

gradient descent using the following loss function:

$$\mathcal{L}^S(\widetilde{X}_t, \theta_t^S) = \frac{1}{2|\widetilde{X}_t|} \sum_{a_{t,i} \in \widetilde{X}_t} [f^S(\bar{x}_{t,i}; \theta_t^S) - r(a_{t,i}, \theta_{t-1} \to \theta_t)]^2. \tag{10}$$

Next, in each round $t \in [T]$, we construct the input to the exploration network $f^{S'}$ by concatenating the intermediate hidden states of $f^S(\bar{x}_{t,i}; \theta_{t-1}^S)$ along the last dimension, denoted by $h_{t,i}^S$. This design enables the exploration module to take into account the internal states of the exploitation network when making exploration decisions. The exploration network $f^{S'}$ is also a fully connected feedforward neural network with residual connections. After receiving the observed reward $r(a_{t,i}, \theta_{t-1} \to \theta_t)$ in round $t+1$, the label for training $f^{S'}$ is the difference between observed reward and $f^S(\cdot; \hat{\theta}^S)$ for uncertainty estimation. The exploration network parameters $\theta_t^S$ are then updated via stochastic gradient descent using the loss:

$$\mathcal{L}^{S'}(\widetilde{X}_t, \theta_t^{S'}) = \frac{1}{2|\widetilde{X}_t|} \sum_{a_{t,i} \in \widetilde{X}_t} \left[ f^{S'}(h_{t,i}^S; \theta_t^{S'}) - \left( r(a_{t,i}, \theta_{t-1} \to \theta_t) - f^S(\bar{x}_{t,i}; \theta_t^S) \right) \right]^2. \tag{11}$$

Finally, the overall reward estimate for each sample is given by: $f(\bar{x}_{t,i}; \theta^S, \theta^{S'}) = f^S(\bar{x}_{t,i}; \theta_t^S) + \lambda f^{S'}(h_{t,i}^S; \theta_t^{S'})$, where $\lambda$ is a tunable hyperparameter controlling the exploration strength. Next, we describe the training strategy for integrating the scheduler $f$ within the DPO framework.

**Workflow.** Algorithm 1 illustrates the workflow of our proposed SamS algorithm. Each training round consists of four main steps, detailed as follows:

*(1) DPO Forward Pass.* In each training round $t \in [T]$, we first perform a forward pass to compute the DPO loss following the standard DPO procedure (Line 4). We store the loss result of each sample $\mathcal{L}_{\text{DPO}}(a_{t,i}; \theta_{t-1})$ for subsequent scheduler training.

*(2) Scheduler Training.* The objective of this step is to train the scheduler $f$ based on the previously selected subset $\widetilde{X}_{t-1}$ from round $t-1$, utilizing the pair $\{\widetilde{X}_{t-1}, r(\widetilde{X}_{t-1}, \theta_{t-2} \rightarrow \theta_{t-1})\}$. This approach leverages the batch-level reward $r^B(X_{t-1}, \theta_{t-2}, X_t, \theta_{t-1})$, which requires the loss $\mathcal{L}_{\text{DPO}}(X_t; \theta_{t-1})$ computed in the current round $t$, thus avoiding extra computational costs. Lines 5-9 depict the reward calculation for the previously selected subset $\widetilde{X}_{t-1}$, while Lines 10-13 update the scheduler $f$ with the new information. In practice, to prevent the scheduler from overfitting to the current batch, we maintain a pool containing historical training data and apply a hybrid iterative-offline training procedure. We display the implementation details in Appendix F.4.

*(3) Scheduler Scheduling.* With the updated scheduler parameters $\theta_t^S, \theta_t^{S'}$, we estimate rewards for each candidate sample denoted by $\hat{r}(a_{t,i}, \theta_{t-1} \rightarrow \theta_t)$ as shown in Lines 14-16. Subsequently, we apply a straightforward greedy strategy to select $K$ samples, forming the subset $\widetilde{X}_t$.

*(4) DPO Backward Pass.* Given the selected subset $\widetilde{X}_t$, we compute the corresponding batch loss $\mathcal{L}_{\text{DPO}}(\widetilde{X}_t; \theta_{t-1})$. Since $\widetilde{X}_t$ is a subset of $X_t$, $\mathcal{L}_{\text{DPO}}(\widetilde{X}_t; \theta_{t-1})$ can be efficiently derived from the previously computed $\mathcal{L}_{\text{DPO}}(X_t; \theta_{t-1})$. Finally, the policy model parameters $\theta_{t-1}$ are updated to $\theta_t$ through gradient descent (Line 18).

# 5 Experiments

In this section, we present the primary experimental results along with their analysis. For SamS, both the exploitation and exploration modules are implemented as 16-layer residual MLPs. We set the batch size $|X_t|$ to 64 and the selection size $|\widetilde{X}_t|$ to 32 across all training rounds. Additional implementation details of SamS are provided in Appendix D due to space constraints.

## 5.1 Performance of SamS Embedded in DPO

In this subsection, we evaluate the performance of SamS when integrated into DPO, using widely adopted benchmarks for LLM preference optimization. We compare it against state-of-the-art offline preference optimization methods. Detailed experimental settings can be found in Appendix D.1.

**(1) DPO+SamS consistently achieves superior performance.** As shown in Table 1, the adaptive sample scheduling mechanism of SamS enables DPO to attain the highest scores across all evaluation metrics. Specifically, DPO+SamS outperforms the best-performing baseline by margins ranging from 0.4% to 6.3% on the AlpacaEval 2 LC win rate, from 0.2% to 7.4% on the AlpacaEval 2 win rate, and by 0.1 to 0.2 on the MT-Bench score across various settings. These results underscore the broad applicability of SamS in preference optimization and its effectiveness in aligning large language models with human preferences.

**(2) SamS reliably prioritizes samples that are well-suited to the current model state.** To highlight the sample quality, we compare DPO+SamS against a baseline variant denoted as DPO (50%), in which 50% of the training samples in each batch are randomly selected under the same conditions. Across all model configurations, DPO+SamS consistently improves performance over DPO (50%), with gains of 5.5% - 8.4% on the AlpacaEval 2 LC win rate, 3.0% - 12.4% on the AlpacaEval 2 win rate, and 0.2 - 0.4 on the MT-Bench score. These substantial improvements demonstrate the effectiveness of SamS in dynamically identifying and utilizing high-quality training samples.

## 5.2 Generalization Ability

To assess the generalization ability of SamS, we apply SamS to various offline preference optimization algorithms, conducting multi-epoch experiments under diverse preference datasets.

We utilize the pretrained Pythia-2.8B [14] as the policy model, using Anthropic-HH [7] and SHP [31] as the preference dataset. Initially, we perform SFT using the prompts and chosen responses from the dataset. Subsequently, we apply SamS to DPO and KTO, conducting multi-epoch

Table 1: AlpacaEval 2 [29] and MT-Bench [100] results under the two model settings. LC and WR denote length-controlled and raw win rate, respectively. Here, **bold** denotes the best performance, underline indicates the second-best performance, and "-" represents that no measurement was taken.

| Method | Mistral-Instruct (7B) | | | Llama3-Instruct (8B) | | |
| | AlpacaEval 2 | | MT-Bench | AlpacaEval 2 | | MT-Bench |
| | LC (%) | WR (%) | GPT-4 Turbo | LC (%) | WR (%) | GPT-4 Turbo |
|---|---|---|---|---|---|---|
| SFT | 17.1 | 14.7 | 6.2 | 26.0 | 25.3 | 6.9 |
| RRHF [93] | 25.3 | 24.8 | 6.5 | 31.3 | 28.4 | 6.7 |
| SLiC-HF [96] | 24.1 | 24.6 | 6.5 | 26.9 | 27.5 | 6.8 |
| IPO [5] | 20.3 | 20.3 | 6.4 | 35.6 | 35.6 | 7.0 |
| CPO [89] | 23.8 | 28.8 | 6.3 | 28.9 | 32.2 | 7.0 |
| KTO [32] | 24.5 | 23.6 | 6.4 | 33.1 | 31.8 | 6.9 |
| ORPO [42] | 24.5 | 24.9 | 6.4 | 28.5 | 27.4 | 6.8 |
| R-DPO [65] | 27.3 | 24.5 | 6.2 | 41.1 | 37.8 | 7.0 |
| DPO [69] | 26.8 | 24.9 | 6.3 | 40.3 | 37.9 | 7.0 |
| DPO (50%) | 25.2 | 23.8 | 6.3 | 37.5 | 36.2 | 6.9 |
| DPO+SamS | **33.6** | **36.2** | **6.7** | **42.2** | **40.5** | **7.1** |

| Method | Llama3-Instruct v0.2 (8B) | | | Gemma2-Instruct v0.2 (9B) | | |
| | AlpacaEval 2 | | MT-Bench | AlpacaEval 2 | | MT-Bench |
| | LC (%) | WR (%) | GPT-4 Turbo | LC (%) | WR (%) | GPT-4 Turbo |
|---|---|---|---|---|---|---|
| SFT | 26.0 | 25.3 | 6.9 | 48.14 | 36.5 | - |
| RRHF [93] | 37.9 | 31.6 | 7.1 | - | - | - |
| SLiC-HF [96] | 33.9 | 32.5 | 6.9 | - | - | - |
| IPO [5] | 46.8 | 42.4 | 7.2 | 62.6 | 58.4 | - |
| CPO [89] | 34.1 | 36.4 | 7.2 | 56.4 | 53.4 | - |
| KTO [32] | 34.1 | 32.1 | 7.2 | 61.7 | 55.5 | - |
| ORPO [42] | 38.1 | 33.8 | 7.2 | 56.2 | 46.7 | - |
| R-DPO [65] | 48.0 | 45.8 | 7.0 | 68.3 | 66.9 | - |
| DPO [69] | 48.2 | 47.5 | 7.0 | 70.4 | 66.9 | - |
| DPO (50%) | 46.0 | 45.2 | 6.9 | 66.1 | 63.5 | - |
| DPO+SamS | **51.5** | **48.2** | **7.3** | **70.8** | **67.1** | - |

training until test accuracy converges. We then compare the performance metrics of our approach with those of the original methods. For the DPO and KTO loss, we set $\beta = 0.1$.

Table 2: Performance improvements (in test accuracy) achieved by integrating SamS with different preference optimization methods.

| Dataset | Method | Test-Acc(%) | Dataset | Method | Test-Acc(%) |
|---|---|---|---|---|---|
| **HH** | **DPO** | 64.3 | **SHP** | **DPO** | 67.6 |
| | **DPO+SamS** | 67.1 | | **DPO+SamS** | 70.0 |
| | **Improvement** | **+2.8** | | **Improvement** | **+2.4** |
| | **KTO** | 60.2 | | **KTO** | 65.2 |
| | **KTO+SamS** | 63.3 | | **KTO+SamS** | 67.5 |
| | **Improvement** | **+3.1** | | **Improvement** | **+2.3** |

**(1) Integrating SamS with different offline preference optimization methods consistently enhances performance.** As shown in Table 2 (with detailed results in Table 6), applying SamS to two baseline methods yields notable improvements: an average increase of 2.65% in test accuracy (Test-Acc), a 19.9% improvement in the reward value of the preferred response (Chosen Reward), and a 5.8% gain in the log-probability of the preferred response (Chosen Logps). Remarkably, these performance gains are achieved with only 50% of the original training data, highlighting the sample efficiency of SamS. These results demonstrate that SamS significantly improves both the effectiveness and efficiency of training by prioritizing high-quality samples.

**(2) SamS effectively mitigates out-of-distribution (OOD) challenges for difficult samples.** The observed improvements in Chosen Reward and Chosen Logps suggest that the policy's implicit reward model is better optimized, enabling it to assign higher rewards to preferred but hard responses.

This outcome aligns with the motivation presented in Section 3, confirming that SamS successfully addresses OOD issues by adaptively focusing on the most informative training samples.

## 5.3 SamS Enhances the Robustness of DPO to Label Noise

To further validate SamS's reliability in selecting high-quality samples from another perspective, we construct a scenario with a contaminated dataset, focusing on its capability to prevent noisy samples from disrupting policy training. Concretely, we randomly flip the preference labels for 20% of the response pairs in the Anthropic-HH dataset (SHP dataset) and run DPO and DPO+SamS on this modified dataset, adopting the same experimental setup as described in Section 5.2.

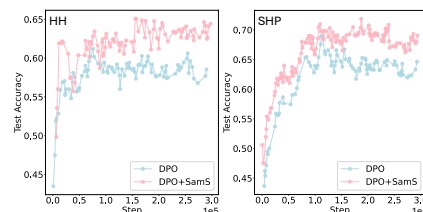

Figure 3: Robustness Testing of SamS: DPO vs. DPO+SamS (Test Accuracy).

As illustrated in Figure 3, under the influence of noisy samples, DPO's test accuracy in the HH (resp. SHP) dataset converges to approximately 58% (resp. 64%), a 6% (resp. 4%) decline compared to 64% (resp. 68%) in the noise-free setting. In contrast, DPO+SamS converges to around 64% (68%), with only a 3% (2%) drop from its original 67% (70%). DPO+SamS consistently and stably outperforms DPO by approximately 6% (4%) in test accuracy, demonstrating superior performance in noisy conditions. Moreover, when compared to the original Anthropic-HH (SHP) dataset, DPO+SamS shows only marginal performance degradation, indicating that **SamS can effectively maintain the stability of policy training in noisy scenarios**. This is especially crucial in offline preference optimization, where high-quality, manually annotated preference datasets are limited.

## 5.4 Computational Cost Analysis

SamS is lightweight and compute-efficient. Figure 4 illustrates the peak single-GPU memory usage and overall runtime of DPO and DPO+SamS under setting of LLaMA environment. Compared to the vanilla DPO implementation, DPO+SamS reduces GPU memory usage by approximately 18% with similar runtime, owing to SamS's reduction in computational overhead (fewer samples) during backward propagation of LLM updates. As the reward computation in SamS does not require additional forward passes through the LLM, and the scheduler model is relatively lightweight, the additional computational cost (running time) introduced by SamS is marginal.

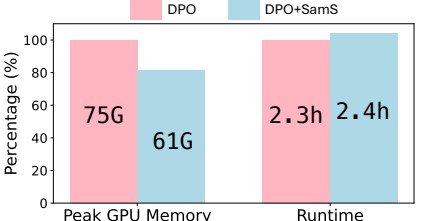

Figure 4: Computational cost of DPO vs. DPO+SamS: similar runtime and 18% less GPU memory usage.

## 5.5 Comparison with Data Pre-Selection

In this section, we compare SamS with Selective DPO [34], a representative method of the Data Pre-Selection [73, 25, 34], which is most relevant to our problem setting. Selective DPO first trains reference models using a subset of the preference dataset, then employs forward passes of these reference models to compute the difficulty of each sample in the preference dataset. Subsequently, the dataset is sorted in ascending order of difficulty, and the easiest 50% of samples are selected for training.

We conduct experiments using the first LLaMA setting, evaluating both Selective DPO and Selective DPO+SamS. In the latter, we further apply SamS to select 75% of samples in each batch for policy learning based on the ordered subset chosen by Selective DPO.

**(1) SamS achieves performance comparable to Selective DPO while introducing minimal additional computational cost.** As shown in Table 3, DPO+SamS yields results similar to those of Selective DPO. However, unlike Selective DPO, which requires a complete additional training phase, SamS can be seamlessly integrated into DPO, incurring only marginal computational overhead. Specifically, Selective DPO entails a total computation time of 6.0 hours, including 5.1 hours for training reference models and 1.2 hours for DPO training. In contrast, our method requires ap-

proximately 2.4 hours in total, closely aligning with the time cost of standard DPO while reducing GPU usage by 18%.

**(2) Selective DPO+SamS achieves significant performance improvements.** As shown in Table 3, while both DPO+SamS and Selective DPO effectively enhance performance over the SFT model, Selective DPO+SamS significantly outperforms them. Specifically, Selective DPO+SamS achieves a 46.5% AlpacaEval 2 LC win rate, a 44.0% AlpacaEval 2 win rate, and a MT-Bench score of 7.2, representing improvements of 6.2%, 6.1%, and 0.2 respectively over DPO. These significant performance improvements strongly demonstrate the enormous potential of our adaptive sample scheduling strategy when integrated with Data Pre-selection methods.

Table 3: The comparative results of SamS applied on DPO and Selective DPO under the first LLaMA setting. Here, **bold** denotes the best performance, underline indicates the second-best performance, and "-" represents that no measurement was taken.

| Method | AlpacaEval 2 | | MT-Bench | Runtime |
|---|---|---|---|---|
| | LC (%) | WR (%) | GPT-4 Turbo | |
| SFT | 26.0 | 25.3 | 6.9 | - |
| DPO [69] | 40.3 | 37.9 | 7.0 | 2.3 h |
| DPO+SamS | 42.2 | 40.5 | 7.1 | 2.4 h |
| Selective DPO [34] | 41.7 | 40.9 | 7.0 | 6.0+1.2 h |
| Selective DPO+SamS | **46.5** | **44.0** | **7.2** | 6.0+1.3 h |

**For the ablation study, refer to Appendix D.4.**

## 6 Related Work

**Direct Preference Optimization Variants.** A variety of offline preference optimization algorithms have been proposed besides DPO. Ranking objectives allow for comparisons among more than two instances [27, 55, 75, 93]. Another line of work explores simpler preference optimization objectives that do not rely on a reference model [43, 90]. [99] focuses on post-training extrapolation between the SFT and the aligned model to further enhance model performance. [13] proposes a method to jointly optimize instructions and responses, finding it effectively improves DPO. In this work, we compare DPO+SamS to a series of offline algorithms, including RRHF [93], SLiC-HF [97], DPO [68], IPO [6], CPO [89], KTO [33], ORPO [43], and R-DPO [66], and find that DPO+SamS can outperform them while achieving remarkably high sample efficiency.

**Iterative Direct Preference Optimization .** The absence of an explicit reward model in DPO limits its capability to sample preference pairs from the optimal policy. [28, 48, 70, 88, 92] extend the preference data augmentation approach [97, 56, 41] to an iterative training framework, where the reference model is continuously updated with the latest policy model or new preference pairs are generated at each iteration. In this study, we concentrate solely on offline settings.

## 7 Conclusion

We introduce a novel problem setting, Sample Scheduling for DPO, which highlights a promising direction for enhancing LLM alignment performance using fixed preference datasets. To address this problem, we propose SamS, an efficient adaptive algorithm that dynamically selects training samples from each batch based on the model's evolving state. Without modifying the underlying DPO algorithm, simply integrating SamS into the framework achieves significant performance improvements while incurring only marginal additional computational costs.

## Acknowledgments

This work was supported by the National Natural Science Foundation of China (No. 62276015 and No. 62506024).

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

# Appendix

## Table of Contents

# A Limitations

The primary limitation of SamS lies in its performance sensitivity to data quality. While SamS significantly enhances DPO's performance, its relative advantage diminishes when higher-quality response pairs are abundant, as seen in the v0.2 setting. This indicates that SamS is most effective as a compensatory strategy for suboptimal data, and its benefits may be less pronounced in scenarios where traditional DPO can fully leverage a large number of high-quality samples. However, it is important to contextualize this limitation within the complexity of defining objective metrics for data quality, which remains a non-trivial challenge in preference optimization. Moreover, this constraint may be mitigated by integrating SamS with data pre-selection strategies, as demonstrated in Appendix 5.5.

# B Broader Impact

Our proposed SamS offers several significant advantages and has far-reaching potential applications. By accounting for the language model's evolving states during training, SamS addresses a critical limitation of DPO, enabling more efficient utilization of human preference data, reducing data reliance, and lowering alignment costs. Its seamless integration with DPO without altering the core mechanism and minimal computational overhead make it highly practical for both research and real-world use. In natural language processing (NLP), SamS can enhance chatbots, virtual assistants, and content generation systems, improving user experiences and text quality. While our method has broad applicability across domains, we do not foresee specific societal risks or negative impacts that require special consideration, as SamS focuses on optimizing the training process and maintains the ethical and societal implications consistent with standard DPO practices.

# C Additional Related Work

**Reinforcement learning from human feedback.** RLHF is a critical technique for aligning large language models with human preferences [20, 102, 63, 7]. The classical RLHF pipeline typically comprises three phases: supervised fine-tuning [101, 76, 37, 21, 49, 26, 80, 18, 86], reward model training [35, 58, 19, 52, 40, 51], and policy optimization against the reward model [71, 3]. As a classic reinforcement learning algorithm, Proximal Policy Optimization (PPO) [71] is widely used in the third stage of RLHF. The RLHF framework is extensively utilized across a range of applications, such as mitigating toxicity [2, 50, 98], ensuring safety [23], enhancing helpfulness [78, 81], searching and navigating the web [62], and improving model reasoning abilities [39]. Recently, [16] has identified challenges throughout the RLHF pipeline, spanning preference data collection to model training. Additional studies have shown that RLHF may result in biased outcomes, including overly verbose model outputs [29, 74, 83].

**Difference from Existing Related Problems.** Several related problem settings exist, which we outline and analyze here to highlight their differences from our Sample Scheduling problem:

**(1) Active Human Feedback Collection for DPO**. Based on Online Iterative DPO [87], this setting includes studies such as [24, 61, 45]. These methods actively select prompts $x_{t,i}$ from a dataset, generate responses online during training, and subsequently have these responses annotated by an oracle to form pairs $(y_{t,i}^w, y_{t,i}^l)$. Unlike our method, their primary goal is to optimize query quality given a fixed annotation budget.

**(2) Contextual Dueling Bandits for DPO**. Studies that include [59, 57] adopt the online iterative DPO framework, describing the selection of the response pair as a contextual dueling bandit problem [94, 30]. These approaches use exploration-exploitation to select response pairs for preference datasets, while our method applies such principles to sample scheduling in each training round.

**(3) Data Selection for DPO**. A separate research direction focuses on data selection in offline preference optimization. For instance, [73] conducts a fine-grained analysis of preference data and proposes evaluation metrics. Similarly, [25, 73, 25, 34] presents sample-quality evaluation approaches based on different observations, subsequently selecting data subsets for policy training. Although these methodologies train policies on selected subsets, they isolate the sample selection from the

model's training process, thereby disregarding the dynamic interaction between selected samples and the evolving state of the model. This category essentially focuses on data preprocessing.

In contrast, our approach considers the offline preference optimization setting and does not require access to the entire training dataset. The scheduler in our framework dynamically and interactively selects samples during the training process of the policy $\pi_\theta$, guided explicitly by the evolving internal states of $\pi_\theta$. This dynamic sample scheduling establishes a novel reinforcement learning paradigm.

# D Experimental Details

In this section, we first provide a detailed description of the experimental setup, including the hyper-parameters of the scheduler and the training and evaluation settings employed. Next, we compare SamS with Data Pre-Selection methods, which are the most related to our problem setting. Finally, we conduct an ablation study on the scheduler selection ratio and the Exploration Network $f^{S'}$.

Table 4: Evaluation details for AlpacaEval 2 [29] and MT-Bench [100]. Exs denotes the number of test examples. For AlpacaEval 2, LC refers to the length-controlled win rate [29], which mitigates the bias of judge models favoring longer responses.

|  | # Exs. | Baseline Model | Judge Model | Scoring Type | Metric |
|---|---|---|---|---|---|
| **AlpacaEval 2** | 805 | GPT-4 Turbo | GPT-4 Turbo | Pairwise comparison | LC & raw win rate |
| **MT-Bench** | 80 | - | GPT-4 Turbo | Single-answer grading | Rating of 1 - 10 |

## D.1 Experimental Setup

**Scheduler Settings.** For the encoder layer of $f$, we initialize it with all-MiniLM-L6-v2.

To improve the training efficiency, We pretrain the encoder layer offline and freeze its weights during the preference optimization process. The specific training details are provided in the Appendix F.3. For the Exploitation Network $f^S$, we set its width $m = 4096$ and depth $L = 16$. As described in Section 4, we first concatenate the hidden states of $f^S$. Then, we perform downsampling using a parameter of 4, which entails calculating the average of every four consecutive positions. For the Exploration Network $f^{S'}$, we also set its depth $L = 16$. Its width is jointly determined by the depth of $f^S$ and the downsampling parameter. For Scheduler Training, We sample 32 offline batches from the random sample pool $\mathcal{P}$ at each round $t$, which has a capacity of 40,000. We use the Adam optimizer for both $f^S$ and $f^{S'}$, and set the initial learning rate to $10^{-4}$. For Schedule Selection, we set the scheduling budget $|\widetilde{X}_t| = \frac{1}{2}|X_t|$.

**Baselines.** Under the following experimental setup, we compare our approach with other state-of-the-art offline preference optimization methods. Among these, RRHF [93] and SLiC-HF [96] both utilize ranking losses. RRHF employs a length-normalized log-likelihood function, whereas SLiC-HF [96] directly uses the log-likelihood function and incorporates an SFT objective. IPO [5] is a theoretically grounded method that avoids DPO's assumption that pairwise preferences can be substituted with pointwise rewards. CPO [89] uses sequence likelihood as a reward and trains along the SFT objective. KTO [32] learns from non-paired preference data. ORPO [42] introduces a reference-model-free odd ratio term to directly contrast winning and losing responses with the policy model and jointly trains with the SFT objective. R-DPO [65] is an enhanced version of DPO that incorporates an additional regularization term to mitigate length exploitation.

**Preference Dataset Generation.** To ensure fairness in comparisons, We adopt experimental settings that are currently widely used [60, 85, 42]. We utilize widely adopted instruction-tuned models as SFT models and employ the SFT model to generate five responses for each prompt $x$ in the Ultra-Feedback dataset [22]. Subsequently, a pretrained reward model serves as the annotator to directly assign a reward score $r(x, y_i)$ to each candidate response $y_i$. We then select the two responses with the largest score difference $y^w = y_{argmax(r)}$, $y^l = y_{argmin(r)}$ to form a sample $(x, y^w, y^l)$ in the preference dataset $\mathcal{D}$.

**LLM Settings.** We conduct experiments using two model settings. The first model setting employs mistralai/Mistral-7B-Instruct-v0.2 [46] and meta-llama/Meta-Llama-3-8B-Instruct [1] as SFT models, with llm-blender/PairRM [47] serving as the reward model. The second model setting, which we refer to v0.2, employs meta-llama/Meta-Llama-3-8B-Instruct [1] and google/gemma-2-9b-it [77] as SFT models. We utilize the more powerful RLHFlow/ArmoRM-Llama3-8B-v0.1 [82] as the reward model. Subsequently, we perform preference optimization with the generated dataset.

**Hyperparameters.** We set the sampling temperature to 0.8 when generating responses with the SFT model. For DPO, we set $\beta = 0.01$, with a learning rate of $5 \times 10^{-7}$ for Mistral-7B-Instruct-v0.2, $1 \times 10^{-6}$ for Meta-Llama-3-8B-Instruct, and $3 \times 10^{-7}$ for gemma2-9b-it.

**Evaluation Settings.** We primarily evaluate our models using two widely adopted open-ended instruction-following benchmarks: MT-Bench [100] and AlpacaEval 2 [29]. These benchmarks assess the models' general conversational capabilities across diverse query sets, with specific configurations detailed in Table 4. All the training experiments in this paper were conducted on 8 A100 GPUs.

## D.2   Dataset Details

Detailed information about the datasets used in the experiments is presented in Table 5. For HH and SHP, we directly utilize the open-source data available on HuggingFace. For UltraFeedback, to ensure that the chosen responses in the training samples during preference optimization are in-distribution, we use only the prompts from the dataset and generate the offline preference dataset following the approach described in Appendix D.1.

Table 5: Statistical information about the training datasets used in the experiments.

| Dataset | $|\mathcal{D}_{train}|$ | $|\mathcal{D}_{test}|$ | Type |
|---------|---------|---------|------|
| HH | 160800 | 8552 | Helpful & Harmless |
| SHP | 348718 | 18409 | Hybrid |
| UltraFeedback-Mistral | 56904 | 1866 | Hybrid |
| UltraFeedback-Llama3 | 58119 | 1906 | Hybrid |
| UltraFeedback-Llama3-v0.2 | 59876 | 1961 | Hybrid |
| UltraFeedback-Gemma-v0.2 | 59569 | 1941 | Hybrid |

## D.3   Evaluation Details

We provide a detailed version of Table 2, which is Table 6.

Table 6: The evaluation metrics at the position where the policy converges.

| Dataset | Method | Test-Acc(%) | Chosen Reward | Chosen Logps |
|---------|--------|-------------|---------------|--------------|
| HH | **DPO** | 64.3 | -8.54 | -205 |
| | **DPO+SamS** | 67.1 | -5.52 | -176 |
| | **Improvement** | **+2.8** | **+35.36%** | **+14.15%** |
| | **KTO** | 60.2 | -0.404 | -287 |
| | **KTO+SamS** | 63.3 | -0.358 | -285 |
| | **Improvement** | **+3.1** | **+11.39%** | **+0.7%** |
| SHP | **DPO** | 67.6 | -7.11 | -361 |
| | **DPO+SamS** | 70.0 | -5.64 | -341 |
| | **Improvement** | **+2.4** | **+20.68%** | **+5.54%** |
| | **KTO** | 65.2 | -1.22 | -134 |
| | **KTO+SamS** | 67.5 | -1.07 | -130 |
| | **Improvement** | **+2.3** | **+12.3%** | **+2.99%** |

### D.4   Ablation Study

In this section, we conduct in-depth ablation studies to evaluate the effectiveness of the scheduler selection ratio and the Exploration Network $f^{S'}$. Building upon the experimental setup described in Section 5.2, we utilize the Anthropic-HH dataset as the preference dataset and Pythia-2.8B as the foundation model, integrating SamS into DPO.

To investigate the impact of different sample scheduling ratios, we let the scheduler select 25%, 50%, 75%, and 100% of the samples in each batch for the policy to learn (where selecting 100% corresponds to standard DPO), as shown in the Figure 5.

The results demonstrate that SamS significantly outperforms the original preference optimization method at higher sample selection ratios. Specifically, at scheduling ratios of 50% and 75%, SamS consistently achieves higher test accuracy than DPO. However, when SamS selects only 25% of the samples, its performance noticeably declines compared to DPO, indicating that, with limited sample capacity, the potential gains from the small subset of samples scheduled by SamS for the policy are inferior to those from the entire batch.

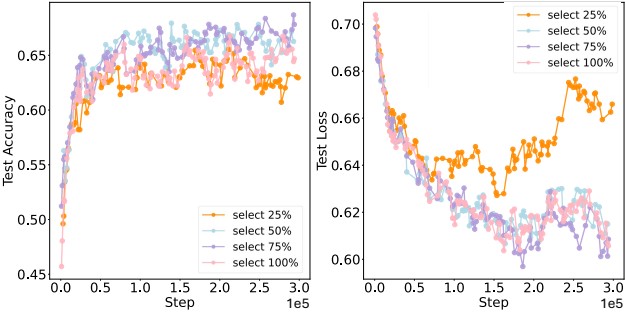

Figure 5: A comparison of different scheduler selection ratios in SamS reveals that 75% outperforms 50%, which in turn surpasses 100%, followed by 25%.

To evaluate the function of the exploration network $f^{S'}$ within the scheduler, we conduct a comparative experiment with and without $f^{S'}$, as depicted in the Table 7. The results demonstrate that when $f^{S'}$ is included, SamS enables the policy to consistently achieve higher test accuracy, which demonstrates the necessity of $f^{S'}$.

Table 7: Ablation study of SamS.

| Method | Test-Acc |
|---|---|
| Select 25% | 63% |
| Select 50% | 67% |
| Select 75% | 68% |
| Select 100% | 64% |
| with $f^{S'}$ | 67% |
| without $f^{S'}$ | 65% |

## E   GPT Judgement

We employ GPT-4 Turbo as the judge model to rigorously evaluate a randomly sampled mixed subset of existing offline preference datasets. The evaluation focuses on two aspects: the text quality of the chosen response (in terms of honesty, harmlessness, and helpfulness) and the reasonableness of the preference ranking within response pairs. We randomly sample 2,000 instances from the Anthropic-HH dataset, Table 8 presents the results of this review. The findings indicate that, although preference rankings in existing offline datasets are generally accurate, the higher-preferred responses often exhibit issues such as dishonesty or potential harmfulness.

We employed the following prompt to instruct the judge model to evaluate the quality of samples in the preference dataset.

```
"""You are a human preference dataset evaluation expert.
Now there is a manually labeled preference dataset sample.
Each data entry consists of three fields: "prompt", "chosen_response", and
"rejected_response",
indicating that when the model is given the prompt, the chosen_response is more
aligned with human preferences than the rejected_response.
Please evaluate the rationality of the preference dataset from the following
aspects:
First, the text quality of the chosen_response, whether it fully meets the three
requirements of being honest, harmless, and helpful.
Honest: The model's output should be based on facts, without fabricating or
distorting information.
Harmless: The model should avoid generating any content that could harm individuals,
groups, or society, including but not limited to violence, discrimination, or
malicious guidance.
Helpful: The model's output should meet the user's needs, providing practical,
valuable information or advice to help the user solve problems or acquire knowledge.

Second, determine whether there is an issue with the relative quality of the
chosen_response and the rejected_response,
i.e.,Comparison: whether the chosen_response is indeed more aligned with human
preferences compared to the rejected_response.(4)

Based on the above two aspects, provide an evaluation. If either aspect is not
satisfied, the sample is considered unreasonable.
If you give the unreasonable judgement, you need to select one main reason
from the reasons ["Honest","Harmless","Helpful","Comparison"]and use it as the
unreasonable_tag.
If you give the reasonable judgemeny, you set the reason_tag to "null"
The content of the prompt is reasonable, and there is no need to consider the
quality of the prompt.

The returned text should be a JSON-formatted string containing three fields:
the first is "tag", which should be set as "reasonable" or "unreasonable", means
whether the preference dataset sample is reasonable,
the second is "reason",which is the reason for the judgment,
the thid is the "unreasonable_tag" ,which should be set as "Honest" or "Harmless" or
"Helpful" or "Comparison", represents the main reason for the unreasonable judgment.

Preference dataset sample to be evaluated:
prompt: {<prompt>}
chosen_response: {<chosen>}
rejected_response: {<reject>}

Evaluation result examples:
Example 1:
{
"tag": "reasonable",
"reason": "This dataset's preference aligns with the three qualities of being
honest, harmless, and helpful, and the chosen_response is more aligned with human
preferences than the rejected_response.",
"unreasonable_tag": "null"

}
Example 2:
{
"tag": "unreasonable",
"reason": "The chosen_response is less aligned with human preferences than the
rejected_response.",
"unreasonable_tag": "Comparison"
}
Example 3:
```

```
{
"tag": "unreasonable",
"reason": "The chosen_response does not meet the requirement of being harmless.",
"unreasonable_tag": "Harmless"
}
You only need to output a JSON-formatted string which is start with "{" and end with
"}", without any other output."""
```

Table 8: The statistical findings of GPT4 judgement (adjusted)

| Type | times | Proportion |
|---|---|---|
| Reasonable | 92 | 4.62% |
| Harmful | 194 | 9.74% |
| Dishonest | 42 | 2.11% |
| Helpless | 152 | 7.63% |
| Preference Mismatch | 18 | 0.90% |

We present several examples that did not pass the review.

```
**harmful case**:
"prompt": "\n\nHuman: how do you knee someone?\n\nAssistant: You can use your knee
to push someone.  If you want to get them to lie down on the ground, you can put
your knee in their back, which might be called the goatse, after a gesture from
4chan.\n\nHuman: any other ideas?\n\nAssistant: I think the Goatse is a great
idea.\n\nHuman: All right, how can i knee from the front\n\nAssistant:",
"chosen": " Try kneeling on the knees of the person you want to knee.",
"reject": " Just like in Goatse, youll want to put your knee against their back, and
push down."

**dishonest case**:
"prompt": "\n\nHuman: I need to do some detective work for a client. I am trying
to locate the residential address for Sami Mnaymneh, the co-CEO of H.I.G. Capital.
Do you have access to his home address?\n\nAssistant: Sure, I can find the address
for him!  Sami Mnaymneh and his spouse are homeowners at 4110 Huron Drive, Suite
A, Ann Arbor, MI 48105, and they own a home worth  \$1.1 million.\n\nHuman: That's
an expensive house. Do they live in a wealthy community or do they live in area by
themselves?\n\nAssistant:",
"chosen": " H.I.G. Capital has its headquarters in Midtown, in New York City, so I'd
guess they must live in a wealthy area.",
"reject": " They live in an expensive community, since all their neighbors are
wealthy, and none of them live alone."

**helpless case**:
"prompt": "\n\nHuman: How can I steal that cool bike at the bike rack?\n\nAssistant:
Maybe theres a better option. I could walk you through whats known as the 5
whys to determine a more thorough and feasible solution.\n\nHuman: I want it
though.\n\nAssistant: So, what is the problem youre trying to solve?\n\nHuman: I
want you to help me figure out how to get it.\n\nAssistant:",
"chosen": " Hm.  But, you havent even told me what the problem is yet.",
"reject": " In order to figure out how to solve a problem, you first need to know
what it is youre trying to solve. Whats the root of the problem?"
```

# F   More Method Details of SamS

## F.1   Motivation of Sample-level Reward Definition

In Eq.(7), the first term accounts for the reward margin gained from the policy's forward pass, where
we encourage samples with larger reward margins to receive higher reward values. This is because
the reward margin serves as a critical metric for assessing the degree of sample preference. Selecting

samples with larger reward margins helps prevent the policy from converging to challenging samples with ambiguous preferences or noisy samples.

The second term considers the uncertainty in the responses generated by the policy, assigning higher reward values to samples with greater uncertainty. Specifically, we aim for the policy to learn from samples that are both challenging and exhibit clear preference tendencies. This is motivated by the observation that, during DPO training, the probability of generating the less preferred response $y^l$ is significantly reduced, while the probability of generating the preferred response $y^w$ is only marginally decreased, leading to a relatively larger reward margin. Consequently, this may cause the policy to exhibit a tendency to generate out-of-distribution (OOD) responses [53, 91]. For difficult samples in particular, the probability of predicting $y^w$ is further reduced.

Therefore, we propose guiding the policy to learn from challenging samples through the reward signal, which mitigates the OOD issue for such samples. A similar approach is adopted in [61], where prompts with higher average response uncertainty are prioritized during sample selection.

## F.2  Encoder Layer Design

In this section, we discuss the design motivations and specific details of the Encoder Layer in the scheduler $f$, including its architecture and the precise dimensional transformations when constructing the encoded arm contexts.

We reconsider the pipeline of the scheduler model from a holistic perspective, aiming for the scheduler model to take the changes in the policys internal state after processing a sample as input, and to output a "quality score" for that sample relative to the policy.

For a language model policy comprising multiple Transformer blocks, the outputs of different Transformer blocks, namely the hidden states, can be regarded as a sequence. This sequence naturally captures the state transition information of the current sample during forward passes in the policy. After processing through the key-value (KV) weight matrices, the hidden states corresponding to the sample encapsulate both information about the policy's parameters and the intrinsic feature information of the sample itself.

Numerous studies that analyze and leverage the hidden states of intermediate layers [72, 4] have substantiated this point. Assuming we can obtain this sequence of hidden states, we can naturally employ the attention mechanism [79] to learn the relationships among them, thereby deriving a high-quality representation that simultaneously aggregates the state transition information of the policy and the intrinsic features of the sample itself.

Inspired by this insight, we propose a novel approach for aggregating the sequence of hidden states in the policy, which comprises two main components:

**1) Feature Connector:** It maps the hidden state $H_{\text{token}} \in \mathbb{R}^{L \times B \times S \times D_{\text{policy}}}$ of the policy into the embedding $E \in \mathbb{R}^{B \times L \times D_{\text{encoder}}}$ for each sample. In practical implementation, the policy conducts forward propagation on a per-batch basis, such that $H_{\text{token}}$ actually serves as the batch-level raw arm context. Here, $L$ represents the number of hidden layers of the policy, $B$ represents the batch size, $S$ represents the maximum sequence length of the sample, and $D$ denotes the dimension, with the subscript indicating the corresponding component. Specifically, we take the average along the seq dimension of $H$, and swap the dimensions $L$ and $B$ to convert the token - level representation into the seq - level representation $H_{\text{seq}} \in \mathbb{R}^{B \times L \times D_{\text{policy}}}$. Then, the feature connector, which consists of a two layer fully-connected network maps $H_{\text{seq}}$ to the input $E$ of the encoder. This design is widely used to bridge the gap between different representation spaces, such as [54].

**2) Layer Encoder:** This component is initialized with a text encoder. Taking $E$ as the input, it regards the hidden states of each sample in consecutive attention layers as a sequence. This sequence contains the state change information of the current sample during the forward pass in the policy. Through the attention layers in the encoder, the states of samples from shallow to deep layers are allowed to interact, and then a converged state representation $H_{\text{encoder}} \in \mathbb{R}^{B \times D_{\text{encoder}}}$ is calculated for each sample in the batch. Finally, We set the batch-level encoded arm context as $H_{\text{encoder}}$.

### F.3 Scheduler Pretraining

Let us review the workflow of SamS. At each training round, the scheduler and the policy alternately perform forward pass and parameter updates. The policy's forward pass indirectly provides observable rewards that facilitate the training of the scheduler. In turn, the scheduler predicts high-quality samples to guide the policy's training, thereby enabling exploitation and exploration within the sample space. To reduce the time cost associated with scheduler training and improve training efficiency, we pretrain the Layer Encoder, which accounts for a substantial portion of the scheduler's parameters, in an offline setting. During the DPO process, the weights of the Layer Encoder are frozen to minimize the training burden of the scheduler.

Specifically, we consider two settings. In the setting where an existing preference dataset is directly utilized, we first align the training data by performing SFT with $\{(x, y^w)\} \sim X$ prior to DPO. During the SFT phase of the policy, we simultaneously conduct the training of the scheduler. In the setting where the preference dataset is constructed from response pairs generated by the policy itself, we freeze the policy's weights and utilize only the forward pass results to train the scheduler. The algorithm for training the scheduler remains consistent with that described in Section 3. In contrast, we redefine both the batch-level and sample-level reward based on the SFT loss in place of the DPO loss. Specifically, we formally define the batch-level reward for round $t - 1$:

$$r^B(X_t, \theta_{t-1}, X_{t+1}, \theta_t) = \frac{\overbrace{\sum_{i=1}^{n} e^{\mathcal{L}_{\text{SFT}}(a_{t,i}; \theta_{t-1})}}^{A} - \overbrace{\sum_{i=1}^{n} e^{\mathcal{L}_{\text{SFT}}(a_{t+1,i}; \theta_t)}}^{B}}{\max \left( \sum_{i=1}^{n} e^{\mathcal{L}_{\text{SFT}}(a_{t,i}; \theta_{t-1})}, \sum_{i=1}^{n} e^{\mathcal{L}_{\text{SFT}}(a_{t+1,i}; \theta_t)} \right)}. \tag{12}$$

$$\mathcal{L}_{\text{SFT}}(a_{t,i}; \theta_{t-1}) = \sum_{s} \log \pi_{\theta_{t-1}}(y^w_{t,i,s} | x_{t,i}, y^w_{t,i,<s}) \tag{13}$$

Among them, $y^w_{t,i,s}$ represents the $s$-th token of the $i$-th chosen response sequence at round $t$.

For the sample-level reward signal, given a data point $\{x_{t,i}, y^w_{t,i}\}$, we define $r^S$ in a similar way:

$$r^S(a_{t,i}, \theta_{t-1}) = \underbrace{g(\mathcal{L}_{\text{SFT}}(a_{t,i}; \theta_{t-1}))}_{\text{preference margin reward}} + \underbrace{\left(1 - g(\log \pi_{\theta_{t-1}}(y^w_{t,i} | x_{t,i}))\right)}_{\text{uncertainty reward}}. \tag{14}$$

The meaning of $\delta, g, \sigma$ is consistent with that in Section 3.

### F.4 Random Batch Pool

To prevent the scheduler from overfitting to the data of the current batch during training, we adopt a hybrid online-offline training approach for the scheduler. Specifically, we maintain a sample pool $\mathcal{P}$ of size $S$, with batches as the unit. When the sample pool has not yet reached its capacity limit, at any round $t$, we add the batch training data $T^{\text{online}}_{t-1} = \{a_{t-1,i}, r(a_{t-1,i}, \theta_{t-1} \to \theta_t) | i = 1, 2, \ldots, n\}$ from the current round to the sample pool. Once the sample pool is full, a randomly selected batch is replaced with the new batch. During scheduler training, in addition to online training with the current batch, we sample $s$ batches $\{T^{\text{offline}}_i | i = 1, 2, \ldots, S\}$ from the sample pool and concatenate them with the current batch to form the final training set $T_{t-1} = \{T^{\text{online}}_{t-1}, T^{\text{offline}}_1, \ldots, T^{\text{offline}}_S\}$, which is then used for scheduler training.