# OpenReview forum: "Adaptive Batch-Wise Sample Scheduling for Direct Preference Optimization"
_NeurIPS.cc/2025/Conference — NeurIPS 2025 poster_

### Official Review · Reviewer_Cv1P · 2025-06-30

**Clarity:** 3
**Significance:** 2
**Originality:** 3
**Rating:** 4
**Confidence:** 4

**Summary:**

This paper introduces SamS (Sample Scheduling for Direct Preference Optimization)​​, a method to dynamically select training samples during DPO based on the evolving states of the language model.
SamS formulates sample scheduling as a contextual bandit problem, with the reward defined by combining batch-level DPO loss improvement and sample level margin & uncertainty.
The SamS model maps LM's hidden states with two output heads, which regressively fit the reward (exploitation network) and reward difference (exploration network) respectively.
Empirically, SamS improves performance on benchmarks like AlpacaEval 2 and enhances robustness to noisy data,

**Questions:**

1. Can the authors compare with more improved baselines (such as SimPO) for both the main experiment (Table 1) and generalization experiment for different learning methods (Table 2)?
2. Can the authors provide ablation study regarding different components of the reward and network design?

**Ethical Concerns:**

["NO or VERY MINOR ethics concerns only"]

**Final Justification:**

The rebuttal has effectively addressed my key concerns, leading me to maintain a positive stance. The authors have successfully incorporated SimPO as a baseline in both main and generalization experiments, significantly strengthening the validity of their results. The additional ablation studies on reward components and network design provide valuable insights into the inner workings of SamS. While the authors have started adapting SamS to GRPO and iterative DPO, the results of these adaptations are not yet presented. However, this does not overshadow the positive contributions of the paper at this stage. Overall, the authors' efforts in addressing the reviewers' comments have improved the quality of the paper, justifying a positive score.

**Limitations:**

The authors have discussed the limitations in their manuscript.

**Quality:**

3

**Strengths And Weaknesses:**

### Strengths
1. Novelty: Introduces a fresh perspective on DPO by focusing on online dynamic sample scheduling.
2. Practicality​​: The contextual bandit setup's reward can be computed straightforwardly during the DPO process, and the lightweight scheduler model also reduces computational overhead. Empirically, the sampled 50% preference data achieves better performance than the full data, further validating the practicality.

### Weaknesses
1. Limited Baselines: Most of the compared learning objectives are generally weaker than DPO in performance, such as IPO, KTO. It remains unclear whether the proposed methods can work with other improved learning objectives or online/iterative DPO variants.
2. Insufficient Ablations: Lack of ablation study on different components of SamS, including the batch/sample-level reward and the exploration/exploitation network design.

---

> ### Author Rebuttal · Authors · 2025-07-31
>
> Thank you for your insightful feedback. In response, we have incorporated SimPO as a baseline in both the main and generalization experiments, as per your suggestion, and conducted a more detailed ablation study, including the individual components of the scheduler’s reward and the exploration network. We hope this addresses your concerns, and we are happy to answer any further questions you may have.
>
> ---
>
> > **W1&Q1: Can the authors compare with more improved baselines (such as SimPO) for both the main experiment (Table 1) and generalization experiment for different learning methods (Table 2)? It remains unclear whether the proposed methods can work with other improved learning objectives or online/iterative DPO variants.**
>
>
> Following your suggestion, we included SimPO as a baseline in Section 5.1 and evaluated SamS's generalization ability on SimPO in Section 5.2.
> For the generalization experiments, we included SamS+SimPO and compared it with SimPO. Specifically, we replaced the DPO loss in the batch-level reward with the SimPO loss, while keeping the rest of the method unchanged. For dataset and model selection, we followed the experimental setup in Section 5.2. Regarding hyperparameters, we adopted the settings recommended in the SimPO paper, namely $β=2.0$ and $γ=1.0$.
>
> | Dataset | Method       | Test-Acc(%) |
> | -------- | ------------ | ----------- |
> | HH       | SimPO        | 57.4%       |
> | HH       | SimPO+SamS   | 61.4%       |
> | HH       | Improvement  | 4.0%        |
> | SHP      | SimPO        | 56.4%       |
> | SHP      | SimPO+SamS   | 61.1%       |
> | SHP      | Improvement  | 4.7%        |
>
> The experimental results are shown in the table above. After deploying SamS on SimPO, we observed accuracy improvements of 4.0% and 4.7% on the HH and SHP datasets, respectively. These improvements are more significant compared to DPO (2.8%/2.4%) and KTO (3.1%/2.3%), demonstrating that SamS can achieve substantial performance gains even on a stronger baseline like SimPO.
> Additionally, we included SimPO as a baseline in Section 5.1.
>
> | Method   | Mistral-instruct (7B)          |       Mistral-instruct (7B)           |    Mistral-instruct (7B)              | Llama3-Instruct (8B)           |       Llama3-Instruct (8B)            |               Llama3-Instruct (8B)    |
> |----------|--------------------------------|------------------|------------------|--------------------------------|------------------|------------------|
> |         | AlpacaEval 2                   |  AlpacaEval 2              |        MT-Bench     | AlpacaEval 2                   | AlpacaEval 2             |        MT-Bench           |
> |          | LC                             | WR               | GPT4 Turbo       | LC                             | WR               | GPT4 Turbo       |
> | SimPO    | 32.1                           | 34.8             | 6.6              | 41.9                           | 38.0             | 7.0              |
> | DPO+SamS | 33.6                           | 36.2             | 6.7              | 42.2                           | 40.5             | 7.1              |
>
>
> | Method   | Llama3-Instruct v0.2 (8B)      |     Llama3-Instruct v0.2 (8B)              |        Llama3-Instruct v0.2 (8B)           | Gemma2-Instruct v0.2 (9B)     |      Gemma2-Instruct v0.2 (9B)            |     Gemma2-Instruct v0.2 (9B)             |
> |----------|--------------------------------|------------------|------------------|--------------------------------|------------------|------------------|
> |          | AlpacaEval 2                   |      AlpacaEval 2        |    MT-Bench              | AlpacaEval 2                   | AlpacaEval 2     |        MT-Bench           |
> |          | LC                             | WR               | GPT4 Turbo       | LC                             | WR               | GPT4 Turbo       |
> | SimPO    | 53.7                           | 47.5             | 7.0              | 72.4                           | 65.0             | -                |
> | DPO+SamS | 51.5                           | 48.2             | 7.0              | 70.8                           | 67.4             | -                |
>
> The experimental results are presented in the table above. It can be observed that our method achieved higher AlpacaEval 2 win rates and MT-Bench scores across four different model settings. DPO+SamS underperformed SimPO in length-controlled win rates on Llama3-Instruct v0.2(8B) and Gemma2-Instruct v0.2(9B), likely due to SimPO’s loss incorporating regularization for sequence length, which makes it particularly strong in length-controlled win rates. However, it is noteworthy that our method achieved superior performance to SimPO in most cases while using only half the number of samples for training.
>
> Additionally, Regarding whether SamS is effective on other preference optimization objectives, the answer is affirmative. We are currently adapting SamS to GRPO and iterative DPO, and have achieved promising results. We invite you to follow our subsequent work for further details.
>
> ---
>
> > **W2&Q2: Insufficient Ablations: Lack of ablation study on different components of SamS, including the batch/sample-level reward and the exploration/exploitation network design.**
>
> We conducted detailed ablation experiments on the individual components of the multi-level reward signal and the exploration network. We followed the experimental setup in Section 5.2, using the HH dataset and Pythia 2.8B as the model.
>
> | Method                     | Test Acc（%） |
> |----------------------------|--------------|
> | SamS                       | 67.1         |
> | SamS without Preference Margin     | 66.2         |
> | SamS without Model Uncertainty     | 66.7         |
> | SamS without Sample Level Reward   | 65.7         |
> | SamS without Batch Level Reward    | 65.1         |
> | SamS without Exploration Network                   | 65.0         |
> | DPO                        | 64.0         |
> | DPO(50%)                   | 62.8         |
>
>
>
> The experimental results are shown in the table above:
>
> - **Batch-level Reward**: When the batch-level reward is removed, the test set accuracy drops by 2.0% compared to SamS. Although all samples within a batch receive the same reward signal, the scheduler can still learn the potential benefits of samples across different batches, which is crucial for enabling exploration and exploitation across the entire sample space.
> - **Sample-level Reward**: When the sample-level reward is removed, the test set accuracy decreases by 1.4% compared to SamS. When we further ablate the preference margin and model uncertainty components of the sample-level reward, the test set accuracy drops by 0.9% and 0.4%, respectively. This indicates that computable metrics play a significant role in reward signal allocation. The preference margin provides more critical guidance in the sample-level reward signal, while the uncertainty measurement may be related to the dataset distribution and the model’s distribution shift.
> - **Exploration Network**: When the exploration network F2 is not used for error estimation, the test set accuracy decreases by 2.1% compared to SamS, highlighting the necessity of retaining this module.This demonstrates the necessity of the exploration network as a substitute for exploration-exploitation algorithms like UCB/TS in addressing the sample scheduling problem, which we modeled as a multi-armed bandit problem.

---

> > ### Comment · Reviewer_Cv1P · 2025-08-06
> >
> > Thank you for your rebuttal. Incorporating SimPO as a baseline and conducting more detailed ablation studies have further enhanced the work. I will maintain a positive score for this submission.

---

> > > ### Author Response · Authors · 2025-08-06
> > > **Response to Reviewer Cv1P**
> > >
> > > Thank you for your timely response and positive feedback. We sincerely appreciate your valuable suggestions, which have significantly strengthened our work by incorporating SimPO as a baseline and conducting more detailed ablation studies. We are happy to address any further questions or comments you may have.

---

### Official Review · Reviewer_5QuT · 2025-07-01

**Clarity:** 2
**Significance:** 2
**Originality:** 3
**Rating:** 4
**Confidence:** 4

**Summary:**

The authors introduce a new problem: sample scheduling in the preference alignment setting and formulate it using the contextual bandit framework. To address this, the authors propose SamS, a scheduling algorithm that adaptively selects training samples from each batch according to the model’s evolving internal states. SamS-augmented preference alignment methods consistently outperform their vanilla counterparts across diverse benchmarks and LLM backbones.

**Questions:**

**Q1:** In Equation (7), how does $g$, a min-max normalization function, map its argument to $[0, 1]$, given that the argument is a scalar? Is the normalization done per batch?

**Q2:** In reference to Appendix D.1, could the authors clarify how the ground-truth batch-level and sample-level rewards are computed during offline scheduler pretraining?

**Q3:** Could the authors comment on the sensitivity of SamS to the hyperparameter $\lambda$, the tunable hyperparameter controlling the exploration strength?

**Q4:** Do the authors observe any notable differences in reward margins between models trained using baseline preference optimization methods and those augmented using SamS during evaluation?

**Ethical Concerns:**

["NO or VERY MINOR ethics concerns only"]

**Final Justification:**

The authors' rebuttal has addressed my concerns. Given the current scarcity of sample scheduling methods in preference learning, I find SamS to be a valuable contribution. Therefore, I maintain my positive rating.

**Limitations:**

Yes.

**Paper Formatting Concerns:**

No.

**Quality:**

3

**Strengths And Weaknesses:**

**Pros:**
1. The paper proposes a novel and underexplored problem: sample scheduling in the context of preference alignment fine-tuning, and formulates it in a contextual bandit framework.
2. The authors propose a novel exploration-exploitation scheduler based on the LLM’s learning feedback to maximize the potential generalization performance.
3. Extensive empirical results show consistent performance gains of SamS across multiple preference-alignment benchmarks.

**Cons:**
1. The batch-level reward in Equation (6) is computed by using the batch-average DPO loss as the estimation for the expected loss across the full distribution. This is a stochastic and potentially biased approximation, and requires further justification or empirical validation.
2. There is a lack of explanation for the initialization of the parameter $\theta\_{1}$ in Algorithm 1.
3. In Appendix D.1, the authors mentioned that they use a frozen pretrained encoder during the preference optimization process; however, this introduces a distributional shift between the context used by the scheduler and the actual learning dynamics of the fine-tuned model, which is not analyzed.

**Minor Typos:**

L102: "Given a subset $\tilde{X\_{t}}\in X\_{t}$ with size K, $|\tilde{X\_{t}}|=$ **k**."

---

> ### Author Rebuttal · Authors · 2025-07-31
>
> Thank you for your valuable feedback. We anwsered each of your detailed concerns below, reporting the model’s performance under different exploration network weights and conducting a comparative analysis of the changes in reward margin gaps after deploying SamS. We hope this resolves your concerns, and we are happy to respond any further questions you may have.
>
> ---
>
> > **W1: The batch-level reward in Equation (6) is computed by using the batch-average DPO loss as the estimation for the expected loss across the full distribution. This is a stochastic and potentially biased approximation, and requires further justification or empirical validation.**
>
> This design is formed by the consideration of trade-off between effectivness and computational cost. Although applying the standard Monte Carlo estimation will induce some bias across distribution, it boost the practicality of Sams without any additional cost to obtain the reward, avoding the exhaustive scan of full dataset. Such essence is analogous in many well known algorithms such as mini-batch SGD and DQN in RL.
> Nevertheless, as the number of training epochs accumulates, such sampling-based estimation will gradually converge to the true expected value, and the bias will accordingly decrease.
>
>
> To verify the effectiveness of batch-level rewards, we have supplemented relevant ablation experiments. We followed the experimental setup in Section 5.2, using the HH dataset and Pythia 2.8B as the model.
>
> | Method                     | Test Acc（%） |
> |----------------------------|--------------|
> | SamS                       | 67.1         |
> | SamS without Batch Level Reward    | 65.1         |
>
> The experimental results are shown in the table above. When the batch-level reward is removed, the test set accuracy drops by 2.0% compared to SamS. Although all samples within a batch receive the same reward signal, the scheduler can still learn the potential benefits of samples across different batches, which is crucial for enabling exploration and exploitation across the entire sample space. In general, this verifies the rationality of constructing batch-level rewards using the Monte Carlo estimation method based on batch-averaged DPO loss.
>
>
> ---
>
> > **W2: There is a lack of explanation for the initialization of the parameter $θ_1$ in Algorithm 1.**
>
> $θ_1$ refers to the parameters of the scheduler at initialization, where the subscript "1" represents the training round. Specifically, since the batch-level reward cannot be calculated when the training round is 0, the scheduler starts training from the second round. Therefore, we mark the initial subscript of the scheduler as 1.
>
> ---
>
> > **W3: In Appendix D.1, the authors mentioned that they use a frozen pretrained encoder during the preference optimization process; however, this introduces a distributional shift between the context used by the scheduler and the actual learning dynamics of the fine-tuned model, which is not analyzed.**
>
> This is very insighful point, and we have considered this issue as well. To address the semantic gap between the text encoder and the policy, we employed a feature connector (Appendix F.2). Prior to preference optimization, we introduced pre-training of the scheduler during the policy’s SFT phase (Appendix F.3), where we opened the full weights of the scheduler, including the text encoder and the feature connector. During the preference optimization phase, although the text encoder is frozen, the connector remains trainable, allowing it to dynamically adapt to the distribution shifts of the policy during training, ensuring further semantic alignment between the pre-trained encoder and the policy. (This approach has been well-established in multimodal domains [1], where a two-layer fully connected layer is used to bridge the semantic gap between visual and text feature spaces, with the visual encoder kept frozen during visual instruction fine-tuning.)
>
> ---
>
> > **Q1:  In Equation (7), how does g, a min-max normalization function, map its argument to [0,1], given that the argument is a scalar? Is the normalization done per batch?**
>
> Yes, your understanding is entirely correct. We transformed the scalar preference margin term into a relative value through min-max normalization within each batch.
>
> ---
>
> > **Q2：Could the authors clarify how the ground-truth batch-level and sample-level rewards are computed during offline scheduler pretraining?**
>
> In Appendix F.3, we provide a detailed description of the scheduler pre-training process, including the construction of the reward signal and the computation of the loss. Specifically, for the batch-level reward, we utilize the batch-level loss gap to measure the potential benefit the policy derives from the training samples, i.e., directly replacing the DPO loss with the SFT loss. For the sample-level reward, we directly use the cross-entropy loss of the SFT loss as the preference margin, meaning the reward margin in the DPO loss is replaced with the negative of the SFT loss, while the model uncertainty term remains unchanged. During preference optimization, we aim for the scheduler to prioritize samples with larger reward margins; thus, we assign higher ground-truth reward values to response-sample pairs with high reward margins. In contrast, during the SFT process, we aim to minimize the gap with the expected response. A smaller SFT loss indicates closer alignment with the expected response, and we want the scheduler to learn to select in-distribution data. Therefore, we assign larger reward values to samples with smaller SFT loss. (We sincerely apologize for omitting the negative sign before $L_{SFT}$ in Equation 14; this will be corrected in subsequent versions.)
>
> ---
>
> > **Q3: Could the authors comment on the sensitivity of SamS to the hyperparameter λ, the tunable hyperparameter controlling the exploration strength?**
>
> We conducted a parameter sensitivity analysis experiment for the exploration network weight λ, as reported. Following the experimental setup in Section 5.2, we used Pythia2.8B as the model and the HH dataset to evaluate SamS performance under different λ values. Specifically, we set the maximum F2 weight to 0.5, as the exploration network fundamentally relies on the error estimation of the network, and its weight should not exceed that of the exploitation network.
>
> | λ    | Test Acc（%） |
> |------|--------------|
> | 0    | 65.0%        |
> | 0.05 | 66.1%        |
> | 0.1  | 67.1%        |
> | 0.2  | 66.4%        |
> | 0.5  | 63.4%        |
>
> The experimental results are presented in the table above. We observed that an exploration ratio of 0.1 is the most suitable, with performance degradation occurring when the exploration ratio is either higher or lower. Specifically, when the exploration ratio reaches 0.5, the accuracy on the test set decreases by 3.7%, indicating that a higher exploration ratio is detrimental to the scheduler’s ability to leverage existing experience for sample scheduling. Conversely, when the exploration ratio is 0.05, a performance drop of 1.0% is observed, suggesting that a lower exploration ratio similarly hinders the scheduler’s ability to select samples with greater potential benefits during continuous preference optimization.
>
> ---
>
> > **Q4: Do the authors observe any notable differences in reward margins between models trained using baseline preference optimization methods and those augmented using SamS during evaluation?**
>
> Under the experimental setup in Section 5.2, we compared the average reward margin on the test set during training for DPO and DPO+SamS. Specifically, we used Pythia2.8B as the model and the HH dataset, calculating the average reward margin on the test set every 100k steps.
>
> | Training Steps | Reward Margin of DPO | Reward Margin of DPO+SamS | Difference |
> | -------------- | -------------------- | ------------------------- | ---------- |
> | 100k           | 0.27                 | 0.31                      | +0.04       |
> | 200k           | 0.42                 | 0.47                      | +0.05       |
> | 300k           | 0.44                 | 0.51                      | +0.07       |
> | 400k           | 0.52                 | 0.61                      | +0.09       |
> | 500k           | 0.61                 | 0.78                      | +0.17       |
>
> As shown in the table above, we observed that DPO+SamS consistently achieves significantly higher reward margins than DPO across different training steps, with this difference becoming more pronounced as training progresses. On one hand, this is due to our sample-level reward, which assigns higher rewards to samples with larger reward margins, encouraging the scheduler to prioritize such samples. On the other hand, through iterative training between the scheduler and the policy, the scheduler’s capability improves, leading to a continuous increase in the quality of selected samples. In contrast, DPO selects samples entirely randomly, resulting in a growing disparity in sample reward margins between the two approaches. In summary, our method consistently and significantly enhances the reward margin during training, validating the effectiveness of SamS.
>
> ---
>
> > **Minor Typos:**
>
> Thank you for pointing out this typo. In fact, what we intend to express is that the size of $X_t$  is $|X_t|=K$, and the size of its subset $\hat{X_t}$ is $|\hat{X_t}|=k$. The expression in L102 is indeed prone to misunderstanding, and we will revise it to this in subsequent versions: "Given a subset $\hat{X_t} \subset X_t$, where $|X_t|=K$, $|\hat{X_t}|=k$". We are very sorry for the impact on your reading experience.
>
>
>
>
> ---
>
> [1] Haotian Liu, Chunyuan Li, Qingyang Wu, and Yong Jae Lee. Visual instruction tuning. Ad1081 vances in neural information processing systems, 36:34892–34916, 2023.

---

> > ### Comment · Reviewer_5QuT · 2025-08-03
> > **Response to Authors**
> >
> > Thanks for the authors' detailed response. I appreciate the authors' efforts in the rebuttal and the additional experimental results provided. I find the justification for the use of batch-level and sample-level rewards convincing (W1, Q2). Overall, many of my concerns have been addressed.

---

> ### Author Response · Authors · 2025-08-03
> **Response to Reviewer 5QuT**
>
> Thank you for your prompt response. We truly appreciate your feedback and are glad to hear that most of your concerns have been addressed. We hope our clarifications and additional results contribute positively to your overall assessment. If you have any further questions or suggestions, we would be more than happy to continue the discussion.

---

### Official Review · Reviewer_ye7p · 2025-07-02

**Clarity:** 2
**Significance:** 2
**Originality:** 2
**Rating:** 4
**Confidence:** 4

**Summary:**

In this paper, the authors propose to schedule sampling  for better DPO. Specifically, the authors formulate sample scheduling for DPO as a contextual bandit problem, and then adaptively select samples from each training batch according to the model's evolving states.

**Questions:**

Why is the generalization experiment in Section 5.2 conducted exclusively on Pythia-2.8B, instead of using the same model family as in Section 5.1?

**Ethical Concerns:**

["NO or VERY MINOR ethics concerns only"]

**Final Justification:**

I have read the rebuttal of the authors, and I adjust my score accordingly.

**Limitations:**

yes

**Quality:**

2

**Strengths And Weaknesses:**

Strengths
1. SamS introduces a new problem: "how to dynamically schedule the preference data during DPO training?" It is the first time to study sample scheduling for DPO.
2. The idea of modeling sample scheduling as a contextual bandit problem is interesting. In this way, it frames the dynamic selection of training samples in DPO as a sequential decision-making process under uncertainty.
3. The proposed algorithm SamS is complementary to current optimization algorithms and can be integrated as a plug-in module in existing DPO-related algorithms.

Weaknesses
1. Considering the cost of sample scheduling, the performance improvement in experimental results is not significant enough. In general, one of my main concerns about sample scheduling is it only improve performance under specific conditions, which may actually hinder the scaling effect when the data volume is sufficiently large. Therefore, to verify the effect of the proposed method, the authors should verify the scaling effect of the proposed method.
2. The authors need to compare with other representative sample scheduling algorithms, rather than only comparing with the vanilla DPO. Because there are also many data filtering or selection algorithms in practice, the authors should also compare them for more convincing results.

---

> ### Author Rebuttal · Authors · 2025-07-31
>
> Thank you very much for your valuable suggestions. In response, we have analyzed the performance of SamS during the gradual scaling of the dataset as per your recommendations, verifying that SamS can still achieve stable and even better performance improvements with large amounts of data. Additionally, we have listed and compared with works related to SamS. Finally, we extended the experiments in Section 5.2 to Llama, validating the generalization ability of SamS across different model families.
>
> ---
>
> >  W1: Considering the cost of sample scheduling, the performance improvement in experimental results is not significant enough. In general, one of my main concerns about sample scheduling is it only improve performance under specific conditions, which may actually hinder the scaling effect when the data volume is sufficiently large. Therefore, to verify the effect of the proposed method, the authors should verify the scaling effect of the proposed method.
>
>
> **In terms of the overhead of sample scheduling**, although our method increases the runtime by a small margin, **it significantly reduces the average GPU memory usage**. In Section 5.4, we verified with Llama3-8B-Instruct that SamS can achieve an **18%** reduction in GPU memory usage at the cost of a **4%** increase in runtime.
> Moreover, as the policy scales, the preference optimization process slows down, and the requirements for computing resources become increasingly stringent. In this scenario, the relative runtime efficiency of SamS will become increasingly higher. This is because, apart from the feature connector (details in Appendix F.2, a two-layer MLP), the size of the scheduler adopted by SamS is fixed, and the overhead for sample scheduling and training is basically constant, achieving an increasingly smaller relative time overhead. On the other hand, since only half of the samples are used, SamS truncates the computation graphs of the discarded samples, thereby reducing memory overhead.
> **In summary, both the time efficiency and space efficiency of SamS improve as the model parameters scales**.
>
> **In terms of SamS's performance,** as a lightweight and plug-and-play method, SamS can achieve performance surpassing that of current mainstream preference optimization methods even when using only 50% of the samples (Section 5.1). Compared with DPO which uses the same training samples, SamS yields an improvement of **5.5% - 8.4%** in the LC win rate on AlpacaEval 2, an improvement of **3.0% - 12.4%** in the win rate on AlpacaEval 2, and an improvement of **0.2 - 0.4** in the MT-Bench score.
>
> **SamS can improve performance under general conditions**, as long as the model encounters varying learning difficulties and data noise—conditions that are commonly observed and discussed in the introduction. By leveraging both batch- and sample-level rewards, SamS learns these patterns and adaptively schedules samples to optimize training. The reward signals are designed to be directly accessible during policy training, without incurring additional computational overhead such as extra forward/backward passes through the LLM or additional dataset scans.
>
> **In cases where the data volume is sufficiently large,** as we mentioned and verified in the introduction, existing preference datasets are unordered and contain noise. When the dataset is scaled up, the same issues persist, and **it amplifies the need for adaptive sample scheduling**, which becomes even more critical in such settings.
> To verify that our method has the ability to scale with increasing data volume, we conducted the following experiment. We adopted the experimental setup from Section 5.2. Within multiple epochs, we fixed the random seed and then gradually increased the number of training samples.
>
> | Training Samples | Test Acc of DPO+SamS(%) | Test Acc of DPO(%)| Difference(%)|
> | -------- | ----------- | ---------- |  ----------    |
> | 200k     | 64.5        | 64.4       |   +0.1        |
> | 400k     | 64.6        | 63.1       |   +1.5        |
> | 600k     | 65.5        | 63.8       |   +1.7        |
> | 800k     | 66.1        | 64.4       |   +1.7        |
> | 1000k    | 66.8        | 64.9       |   +1.9        |
>
> As shown in the table above, the performance gap between DPO+SamS and DPO is gradually widening. In other words, our scheduler continues to learn during the training process, and its capability becomes increasingly stronger as the number of training samples increases. This ensures that our method has the ability to scale with larger data volumes.
>
> ---
>
> > W2：The authors need to compare with other representative sample scheduling algorithms, rather than only comparing with the vanilla DPO. Because there are also many data filtering or selection algorithms in practice, the authors should also compare them for more convincing results.
>
> Thanks for the reviewer's insightful suggestions. **We in fact included the related work discussion in Appendix C and the comparison with a representative data selection algorithm in Appendix D.4 in the manuscript (Supplementary Material)**. For ease of reference, we re-present the discussion and results below.
>
> **Related work**.  "Data Selection for DPO" is directly relevant to our problem setting. Specifically, [1] conducts a fine-grained analysis of preference data and proposes evaluation metrics. Similarly, [2,3] presents sample-quality evaluation approaches based on different observations, subsequently selecting data subsets for policy training. Although these methodologies train policies on selected subsets, they isolate the sample selection from the model’s training process, thereby disregarding the dynamic interaction between selected samples and the evolving state of the model. This category essentially focuses on data preprocessing.
>
> **Experiments**. We regret to note that among these works, only Selective DPO [3] has been open-sourced. We conducted experiments using the first LLaMA setting in comparison with Selective DPO, evaluating both Selective DPO and Selective DPO+SamS.
>
> | Method               | AlpacaEval 2 |   AlpacaEval 2        | MT-Bench       | Runtime   |
> |----------------------|--------------|----------|----------------|-----------|
> |                      | LC (%)       | WR (%)   | GPT-4 Turbo    |           |
> | SFT                  | 26.0         | 25.3     | 6.9            | -         |
> | DPO          | 40.3         | 37.9     | 7.0            | 2.3 h     |
> | DPO+SamS             | 42.2         | 40.5     | 7.1            | **2.4 h**     |
> | Selective DPO   | 41.7         | 40.9     | 7.0            | **6.0+1.2 h** |
> | Selective DPO+SamS   | 46.5         | 44.0     | 7.2            | 6.0+1.3 h |
>
> As is shown in the table above, Selective DPO entails a total computation time of 6.0 hours, including 5.1 hours for training reference models and 1.2 hours for DPO training. In contrast, our method requires approximately 2.4 hours in total, closely aligning with the time cost of standard DPO while reducing GPU usage by 18%. This demonstrates that **SamS achieves comparable performance while being far more time-efficient than Selective DPO**.
>
> Furthermore, we found that when combined with Selective DPO, using only 25% of the sample size of the original DPO yields significantly better performance. **Selective DPO+SamS** achieves a 46.5% AlpacaEval 2 LC win rate, a 44.0% AlpacaEval 2 win rate, and a MT-Bench score of 7.2, **representing improvements of 6.2%, 6.1%, and 0.2 respectively over DPO**. This strongly validates the research value of the sample scheduling problem we defined, as well as the enormous application potential of our adaptive sample scheduling strategy when integrated with data pre-selection methods.
>
> ---
>
> > Q1:Why is the generalization experiment in Section 5.2 conducted exclusively on Pythia-2.8B, instead of using the same model family as in Section 5.1?
>
> Thank you very much for your suggestions. Since we followed the experimental setup of DPO, the model family used in our experiments is also Pythia, consistent with that in DPO. In Section 5.1, we used models from other families mainly to verify the performance of our method under currently widely adopted experimental setups in a fair manner.
> As suggested by the reviewer,, we replace the base model in Section 5.2 with **Llama-3-8b (base)** to conduct the following generalization performance experiments.
>
> | Dataset | Method       | Test-Acc(%) |
> | -------- | ------------ | ----------- |
> | HH       | DPO          | 67.1%       |
> | HH        | DPO+SamS     | 69.5%       |
> | HH       | Improvement  | 2.4%        |
> | SHP      | DPO          | 76.9%       |
> | SHP        | DPO+SamS     | 81.6%       |
> | SHP         | Improvement  | 4.7%        |
>
> The experimental results are shown in the table above. Under both the HH and SHP preference datasets, significant performance improvements were achieved after deploying SamS on DPO. This indicates that SamS can effectively generalize to models from other families.
>
> ---
>
> [1]Judy Hanwen Shen, Archit Sharma, and Jun Qin. Towards data-centric rlhf: Simple metrics for preference dataset comparison. arXiv preprint arXiv:2409.09603, 2024.
>
> [2]Xun Deng, Han Zhong, Rui Ai, Fuli Feng, Zheng Wang, and Xiangnan He. Less is more: Improving llm alignment via preference data selection. arXiv preprint arXiv:2502.14560, 2025.
>
> [3]Chengqian Gao, Haonan Li, Liu Liu, Zeke Xie, Peilin Zhao, and Zhiqiang Xu. Principled data selection for alignment: The hidden risks of difficult examples. arXiv preprint arXiv:2502.09650, 2025.

---

> ### Author Response · Authors · 2025-08-05
> **Response to Reviewer ye7p**
>
> We truly appreciate the time and effort you’ve dedicated to reviewing our work. As the discussion period draws to a close, could you kindly let us know whether our clarifications and new experimental results have helped address your concerns? We truly appreciate your time and consideration.

---

### Official Review · Reviewer_Z1FM · 2025-07-18

**Clarity:** 3
**Significance:** 3
**Originality:** 3
**Rating:** 5
**Confidence:** 4

**Summary:**

The paper proposes SamS, an adaptive preference data scheduling algorithm for DPO, which leads to increased winrate performance. SamS is formalized as a contextual bandit problem where at each round the scheduler chooses a subset of the available data to train on. The reward is a linear combination of two terms -- one for the batch-level reward based on the normalized improvement on the exponentiated DPO loss before and after training on the batch, and a sample-level reward to prefer comparisons with large preference margins and model uncertainty. The rewards are then sed to train a scheduler which predicts the reward of each sample based on the previous round, as well as the expected squared error of the reward prediction vs the actual reward. Both are trained with MSE losses. This allows the scheduler to greedily select high value rewards with an optimism term based on the uncertainty.

The paper then tests the effectiveness of SamS in finetuning a variety of models for AlpacaEval 2 and MT-Bench. SamS proves effective at winrate, length controlled winrate, and MT-Bench, outperforming DPO with both the same amount of data, as well as using twice as many (unprioritized) datapoints. This shows that SamS is effective at prioritizing data, even if this results in using less data. SamS also improves performance of KTO on helpful/harmless and SHP. SamS also improves the robustness of DPO to label noise, as shown in Figure 3's ablation which flips 20% of labels.

**Questions:**

1) It could be nice to see ablations for the different terms in the reward. What happens if we don't look at the context level reward? What if we drop the model uncertainty reward? The classification level
2) In the label noise experiment, does SamS tend to discard comparisons with flipped labels?
3) Do you have more hypotheses about why SamS works? Is it also effective in other supervised classification tasks?
4) Is SamS effective at smaller and larger model sizes as well? My guess would be yes, but it's good to check.

**Ethical Concerns:**

["NO or VERY MINOR ethics concerns only"]

**Final Justification:**

The authors added more ablations and model sizes, increasing my confidence in their results. I would have been more convinced if they had showed that their method works on other supervised learning settings, but I still think this paper should be accepted.

I maintain my score of 5, but increase my confidence from 3 to 4.

**Limitations:**

yes

**Paper Formatting Concerns:**

no concerns

**Quality:**

3

**Strengths And Weaknesses:**

Strengths:

Quality: The paper supports its main conclusions with clear experimental results.
Clarity: The paper is generally clearly written.
Significance: The proposed method is competitive against standard methods in the field, including the most relevant baselines such as simply not using the scheduler and instead training on all of the data. Data quality and label noise are important in the RLHF setting, and SamS helps improve the robustness of trianing.
Originality: As far as I am aware, the work is original.

Weaknesses:
W1) A minor point is that the paper only looks at models in the 7-9b parameter scale. It analyzes a variety of models, but it could be nice to see smaller and bigger models as well.
W2) It could be nice to see more experiments + analysis for why SamS works, but I find the paper fairly convincing overall.
W3) (nit) The paper could define/explain MT-Bench (GPT-4 Turbo) in the body of the paper rather than relying on the external citation.

---

> ### Author Rebuttal · Authors · 2025-07-31
>
> Thank you for your valuable feedback. We anwsered each of your detailed questions and supplemented our work with more comprehensive ablation studies as per your suggestions. We hope these responses resolve your concerns, and we are happy to respond any further inquiries you may have.
>
> ---
>
> >**W1/Q4: Is SamS effective at smaller and larger model sizes as well? My guess would be yes, but it's good to check.**
>
> Thank you for your insightful question. Yes, as long as the model is affected by data quality, SamS can learn error patterns and avoid such samples, which is not directly related to the model’s parameter size.
> For your suggestions of evaluating SamS’s performance across models of varying sizes, due to hardware constraints, we were unable to experiment with larger models. On the other hand, for models with fewer than 7B parameters, most are around 3B parameters. Due to their limited parameter size, datasets like AlpacaEval are overly challenging for these models (as evidenced by the AlpacaEval leaderboard, where nearly all models have parameters exceeding 7B). Ultimately, we selected lxuechen/phi-2-sft as the small model for our main experiments, as it is one of the few SFT models that performs exceptionally well on AlpacaEval. We set the learning rate of Phi-2 to 5e-7, and the other hyperparameters remain consistent with those in Appendix D.1.
>
>  | Method      | AlpacaEval 2 LC(%) | AlpacaEval 2 WR(%） |
> |-------------|-----------------------------|-----------------------------|
> | SFT         | 5.9                         | 4.0                         |
> | DPO         | 9.0                         | 8.5                         |
> | DPO(50%)    | 8.3                         | 8.0                         |
> | DPO+SamS    | 10.3                        | 10.2                        |
>
> The table above presents our experimental results, demonstrating that our method consistently outperforms DPO under the small model experimental setup, validating SamS’s generalization capability across models of different sizes. However, even after preference optimization, the absolute win rate remains low, primarily due to the limited parameter size of the model, which makes it challenging to achieve the strong conversational abilities observed in larger models.
>
> ---
>
> > **W2: It could be nice to see more experiments + analysis for why SamS works, but I find the paper fairly convincing overall.**
>
>
> Thank you for your interest in the working principles of SamS. Sams can improve the policy's performance, as long as the policy encounters varying learning difficulties and data noise—conditions that are commonly observed and discussed in the introduction. By leveraging both batch- and sample-level rewards, Sams learns these patterns and adaptively schedules samples to optimize training according to relative difficulty and filtering out the noise data.
>
> Regarding the design of the reward signal, we have supplemented more detailed ablation experiments, which you can find in the detailed analysis provided in the response to Q1. This set of ablation experiments well explains how the reward signal guides the scheduler to work.
>
>
> ---
>
> > **W3: The paper could define/explain MT-Bench (GPT-4 Turbo) in the body of the paper rather than relying on the external citation.**
>
> Thank you very much for your suggestions, and we sincerely apologize for any inconvenience caused by your reading. We have added an explanation for GPT-4 Turbo in the caption of Table 1. The revised caption is as follows:
>
> Table 1: AlpacaEval 2 [26] and MT-Bench [96] results under the two model settings. LC and WR denote length-controlled and raw win rate, respectively. **GPT-4 Turbo refers to the evaluation scores of MT-Bench obtained using GPT-4 Turbo as the judge model.** Here, bold denotes the best performance, underline indicates the second-best performance, and "-" represents that no measurement was taken.
>
> ---
>
> > **Q1: It could be nice to see ablations for the different terms in the reward. What happens if we don't look at the context level reward? What if we drop the model uncertainty reward?**
>
> We conducted detailed ablation experiments on the individual components of the multi-level reward signal and the exploration network. We followed the experimental setup in Section 5.2, using the HH dataset and Pythia 2.8B as the model.
>
> | Method                     | Test Acc（%） |
> |----------------------------|--------------|
> | SamS                       | 67.1         |
> | SamS without Preference Margin     | 66.2         |
> | SamS without Model Uncertainty     | 66.7         |
> | SamS without Sample Level Reward   | 65.7         |
> | SamS without Batch Level Reward    | 65.1         |
> | SamS without Exploration Network                   | 65.0         |
> | DPO                        | 64.0         |
> | DPO(50%)                   | 62.8         |
>
>
>
> The experimental results are shown in the table above:
>
> - **Batch-level Reward**: When the batch-level reward is removed, the test set accuracy drops by 2.0% compared to SamS. Although all samples within a batch receive the same reward signal, the scheduler can still learn the potential benefits of samples across different batches, which is crucial for enabling exploration and exploitation across the entire sample space.
> - **Sample-level Reward**: When the sample-level reward is removed, the test set accuracy decreases by 1.4% compared to SamS. When we further ablate the preference margin and model uncertainty components of the sample-level reward, the test set accuracy drops by 0.9% and 0.4%, respectively. This indicates that computable metrics play a significant role in reward signal allocation. The preference margin provides more critical guidance in the sample-level reward signal, while the uncertainty measurement may be related to the dataset distribution and the model’s distribution shift.
> - **Exploration Network**: When the exploration network F2 is not used for error estimation, the test set accuracy decreases by 2.1% compared to SamS, highlighting the necessity of retaining this module.This demonstrates the necessity of the exploration network as a substitute for exploration-exploitation algorithms like UCB/TS in addressing the sample scheduling problem, which we modeled as a multi-armed bandit problem.
>
> ---
>
> > **Q2: In the label noise experiment, does SamS tend to discard comparisons with flipped labels?**
>
> Yes, Sams tends to discard samples with flipped labels. Sams is an adaptive and dynamic sample selection algorithm. When it encounters samples with flipped labels, it receives negative rewards, as such samples degrade the generalization performance of the policy. Through these interactions, Sams learns to identify such patterns and gradually avoids selecting flipped samples in subsequent training rounds. Ultimately, Sams enhances the robustness of the learned policies or LLMs.
>
> ---
>
> > **Q3: Do you have more hypotheses about why SamS works? Is it also effective in other supervised classification tasks?**
>
>
> The hypotheses we have for Sams is based on the presence of noise in the dataset and the varying learning difficulty of samples. As mentioned in the introduction, SamS can adaptively select samples that maximize the potential gains for the policy when faced with different levels of difficulty.
>
> Moreover, we can model all training data selection problems in a similar manner (Section 3). To the best of our knowledge, this is the first work to model sample selection during training as a multi-armed bandit problem. Specifically, by determining the arm context for each sample in the multi-armed bandit framework and constructing effective true rewards, our algorithmic framework can be applied to sample selection. As illustrated in Appendix F.3, we constructed the reward signal for supervised instruction fine-tuning in a similar manner. However, we have currently conducted a comprehensive exploration only on offline direct preference optimization. This is primarily because obtaining explicit human preference annotations is challenging, and compared to other types of datasets, such as those for supervised instruction fine-tuning, preference datasets exhibit more significant noise, making data selection particularly valuable.

---

> > ### Comment · Reviewer_Z1FM · 2025-08-08
> >
> > W1/Q4: Thanks for the scaling experiment! I agree that testing at smaller model scales is helpful for showing that results are likely to hold over a variety of model scales, and am satisfied with the Phi2 experiments.
> >
> > W3: Thank you for the clarification.
> >
> > W2/Q1: Thanks for the ablations! It seems that all of the components of SamS are necessary to unlock full performance, though it would be helpful to also add error bars.
> >
> > ----
> >
> > One quick (but probably not score-changing) clarification on Q2 -- is there a quantitative measure of how often SamS throws out incorrect labels?
> >
> > Q3: I think that checking on standard classification settings is outside of the original motivation of the work, but would show that the underlying principles of SamS apply in other settings, and are more likely to work in other contexts.
> >
> > ---
> >
> > Given the responses, I feel more confident in my recommendation to accept the paper.

---

> > > ### Author Response · Authors · 2025-08-09
> > > **Response to Reviewer Z1FM**
> > >
> > > Thank you for your recognition of our work!
> > >
> > > ---
> > >
> > > Regarding Q2, we conducted an urgent discussion and propose a feasible approach: constructing an index for the preference dataset along with a marker indicating whether a label is flipped. During training, this marker enables us to track whether SamS effectively discards a given sample. We conducted experiments under a 0.2 noise environment, following the experimental setup in Section 5.2, using the Pythia-2.8b model and the HH dataset with 20% of labels flipped. In the third epoch of training (as DPO had not converged in the first epoch, and SamS had already tend toward exploitation behaviors, achieving relatively stable performance; due to time constraints, we directly loaded the checkpoint after two epochs of training, including the policy and scheduler), processing each sample once, and used an additional marker to indicate whether SamS discarded the sample. By comparing the consistency of these two markers, we evaluated SamS's ability to identify flipped samples. The consistency between the markers was calculated and reported as the accuracy of SamS in discarding samples, as shown in the table below:
> > >
> > > | Method          | Drop Acc |
> > > |-----------------|----------|
> > > | DPO+SamS        | 76.4%    |
> > >
> > > The results demonstrate that SamS can effectively identify and discard flipped samples.
> > >
> > > ---
> > >
> > > For Q3, our future work will focus on applying this sample selection framework to additional supervised training tasks, and we look forward to your continued interest.
> > >
> > > ---
> > >
> > > Once again, we sincerely appreciate your recognition of our work and are happy to address any further questions or comments you may have.

---

### Note · Authors · 2025-08-12

Dear Reviewers, ACs, SACs, and PCs,

We sincerely appreciate your time and effort in evaluating our paper.

---
We are grateful to the reviewers for recognizing **the value of our work in the following aspects**:

- **Novel Problem (Reviewers 5QuT, ye7p, Z1FM, Cv1P)**: Introducing adaptive sample scheduling for direct preference optimization, with the potential to extend to broader RL and supervised learning scenarios
- **New Methodology (Reviewers Z1FM, 5QuT, ye7p)**: Formulating the problem as a contextual bandit and proposing a lightweight exploration–exploitation algorithm SamS
- **Strong Empirical Evidence (Reviewers Cv1P, Z1FM, 5QuT)**: SamS delivers consistent performance improvements across multiple benchmarks, with minimal additional computational overhead and reduced memory consumption

---

In the rebuttal, we have provided the following clarifications and additional results.

### Clarifications on Paper Details
- Explanation of the principles and concepts underlying SamS, description of the reward signal calculation, enhanced related work discussion, and analysis of time and space efficiency

### Additional Experimental Results
- **Ablation Studies** for each reward component (Reviewers Z1FM(Q1), Cv1P(W2, Q2), 5QuT(W1))
- **Performance Robustness** (Reviewers ye7p(W1, Q1), Z1FM(W1, Q4)): Demonstrating that SamS maintains stable performance across different dataset sizes, model parameter scales, and model families
- **New Baselines** (Reviewers Cv1P(W1, Q1), ye7p(W2)): Adding SimPO and Selective DPO for comparison
- **Noise Experiments** (Reviewer Z1FM(Q2)): Showing that SamS correctly identifies and discards flipped samples
- **Reward Margin Evaluation** (Reviewer 5QuT(Q4)): Verifying that our method continuously and significantly improves the reward margin during training compared to DPO

We greatly appreciate the reviewers’ valuable suggestions, which have undoubtedly strengthened this work. We will definitely incorporate them into the revised version.

---

**Dear Reviewer ye7p,**

We respectfully request your feedback again and sincerely invite you to review our responses to the other reviewers as well. **We have dedicated over 60 hours to preparing this rebuttal, and we truly hope our responses address your concerns. If you have any further questions or concerns, we would be glad to address them.**

---

Thank you once again for your time and invaluable insights.

Best regards,

The authors of submission 11557

---

### Decision · Program_Chairs · 2025-09-17

**Decision:**

Accept (poster)

**Comment:**

The paper introduces the problem of Sample Scheduling for DPO (SamS), focusing on dynamically selecting training samples during DPO based on the evolving state of the language model. SamS formulates the sample scheduling problem as a contextual bandit problem, allowing for efficient and adaptive sample selection. The reward function combines batch-level DPO loss improvement and sample-level margin & uncertainty, enabling the scheduler to learn and prioritize valuable samples. The proposed algorithm demonstrates consistent performance gains across various benchmarks and large language model backbones.

During the discussion phase, the reviewers raised a few concerns regarding the significance of performance improvements, comparisons against other methods, and abalation studies. The authors have effectively addressed the reviewers’ concerns, providing additional experiments and clarifications that strengthen the paper’s claims.

In the end, all reviewers agreed that the technical contributions in this work are significant, and the proposed method demonstrates clear benefits in terms of performance and robustness. Therefore, we recommend acceptance of this submission.